# Antibody evasion by SARS-CoV-2 Omicron subvariants BA.2.12.1, BA.4 and BA.5

Qian Wang[1,5], Yicheng Guo[1,5], Sho Iketani[1,2], Manoj S. Nair[1], Zhiteng Li[1], Hiroshi Mohri[1], Maple Wang[1], Jian Yu[1], Anthony D. Bowen[1,3], Jennifer Y. Chang[3], Jayesh G. Shah[3], Nadia Nguyen[1], Zhiwei Chen[4], Kathrine Meyers[1,3], Michael T. Yin[1,3], Magdalena E. Sobieszczyk[1,3], Zizhang Sheng[1], Yaoxing Huang[1], Lihong Liu[1✉] & David D. Ho[1,2,3✉]

SARS-CoV-2 Omicron subvariants BA.2.12.1 and BA.4/5 have surged notably to become dominant in the United States and South Africa, respectively[1,2]. These new subvariants carrying further mutations in their spike proteins raise concerns that they may further evade neutralizing antibodies, thereby further compromising the efficacy of COVID-19 vaccines and therapeutic monoclonals. We now report findings from a systematic antigenic analysis of these surging Omicron subvariants. BA.2.12.1 is only modestly (1.8-fold) more resistant to sera from vaccinated and boosted individuals than BA.2. However, BA.4/5 is substantially (4.2-fold) more resistant and thus more likely to lead to vaccine breakthrough infections. Mutation at spike residue L452 found in both BA.2.12.1 and BA.4/5 facilitates escape from some antibodies directed to the so-called class 2 and 3 regions of the receptor-binding domain[3]. The F486V mutation found in BA.4/5 facilitates escape from certain class 1 and 2 antibodies but compromises the spike affinity for the viral receptor. The R493Q reversion mutation, however, restores receptor affinity and consequently the fitness of BA.4/5. Among therapeutic antibodies authorized for clinical use, only bebtelovimab retains full potency against both BA.2.12.1 and BA.4/5. The Omicron lineage of SARS-CoV-2 continues to evolve, successively yielding subvariants that are not only more transmissible but also more evasive to antibodies.

Severe acute respiratory syndrome-coronavirus-2 (SARS-CoV-2) Omicron or B.1.1.529 variant continues to dominate the coronavirus disease-2019 (COVID-19) pandemic. Globally, the BA.2 subvariant has rapidly replaced previous subvariants BA.1 and BA.1.1 (Fig. 1a). The recent detection and notable expansion of three new Omicron subvariants have raised concerns. BA.2.12.1 emerged in the United States in early February and expanded substantially (Fig. 1a), now accounting for over 55% of all new SARS-CoV-2 infections in the country[2]. BA.4 and BA.5 emerged in South Africa in January and rapidly became dominant there with a combined frequency of over 88% (ref. [4]). These new Omicron subvariants have been detected worldwide, with a combined frequency of over 50% in recent weeks. However, their growth trajectories in the United States and South Africa indicate a substantial transmission advantage that will probably result in further expansion, as is being observed in countries such as the United Kingdom (Fig. 1a). Phylogenetically, these new subvariants evolved independently from BA.2 (Fig. 1b). The spike protein of BA.2.12.1 contains L452Q and S704L alterations in addition to the known mutations in BA.2, whereas the spike proteins of BA.4 and BA.5 are identical, each with four more alterations: Del69-70, L452R, F486V and R493Q, a reversion mutation (Fig. 1c). The location of several of these mutations within the receptor-binding domain (RBD) of

the spike protein raises the spectre that BA.2.12.1 and BA.4/5 may have evolved to further escape from neutralizing antibodies.

## Neutralization by monoclonal antibodies

To understand antigenic differences of BA.2.12.1 and BA.4/5 from previous Omicron subvariants (BA.1, BA.1.1 and BA.2) and the wild-type SARS-CoV-2 (D614G), we produced each pseudovirus and then assessed the sensitivity of each pseudovirus to neutralization by a panel of 21 monoclonal antibodies (mAbs) directed to known neutralizing epitopes on the viral spike. Among these, 19 target the four epitope classes in the RBD[3], including REGN10987 (imdevimab)[5], REGN10933 (casirivimab)[5], COV2-2196 (tixagevimab)[6], COV2-2130 (cilgavimab)[6], LY-CoV555 (bamlanivimab)[7], CB6 (etesevimab)[8], Brii-196 (amubarvimab)[9], Brii-198 (romlusevimab)[9], S309 (sotrovimab)[10], LY-CoV1404 (bebtelovimab)[11], ADG-2 (ref. [12]), DH1047 (ref. [13]), S2X259 (ref. [14]), CAB-A17 (ref. [15]) and ZCB11 (ref. [16]), as well as 1–20, 2–15, 2–7 (ref. [17]) and 10–40 (ref. [18]) from our group. Two other mAbs, 4–18 and 5–7 (ref. [17]), target the N-terminal domain (NTD). Our findings are shown in Fig. 2a, as well as in Extended Data Fig. 1 and Extended Data Table 1. Overall, 18 and 19 mAbs lost neutralizing activity completely or partially against BA.2.12.1 and BA.4/5, respectively.

[1]Aaron Diamond AIDS Research Center, Columbia University Vagelos College of Physicians and Surgeons, New York, NY, USA. [2]Department of Microbiology and Immunology, Columbia University Vagelos College of Physicians and Surgeons, New York, NY, USA. [3]Division of Infectious Diseases, Department of Medicine, Columbia University Vagelos College of Physicians and Surgeons, New York, NY, USA. [4]AIDS Institute and Department of Microbiology, Li Ka Shing Faculty of Medicine, The University of Hong Kong, Pokfulam, Hong Kong Special Administrative Region, China. [5]These authors contributed equally: Qian Wang, Yicheng Guo. ✉e-mail: ll3411@cumc.columbia.edu; dh2994@cumc.columbia.edu

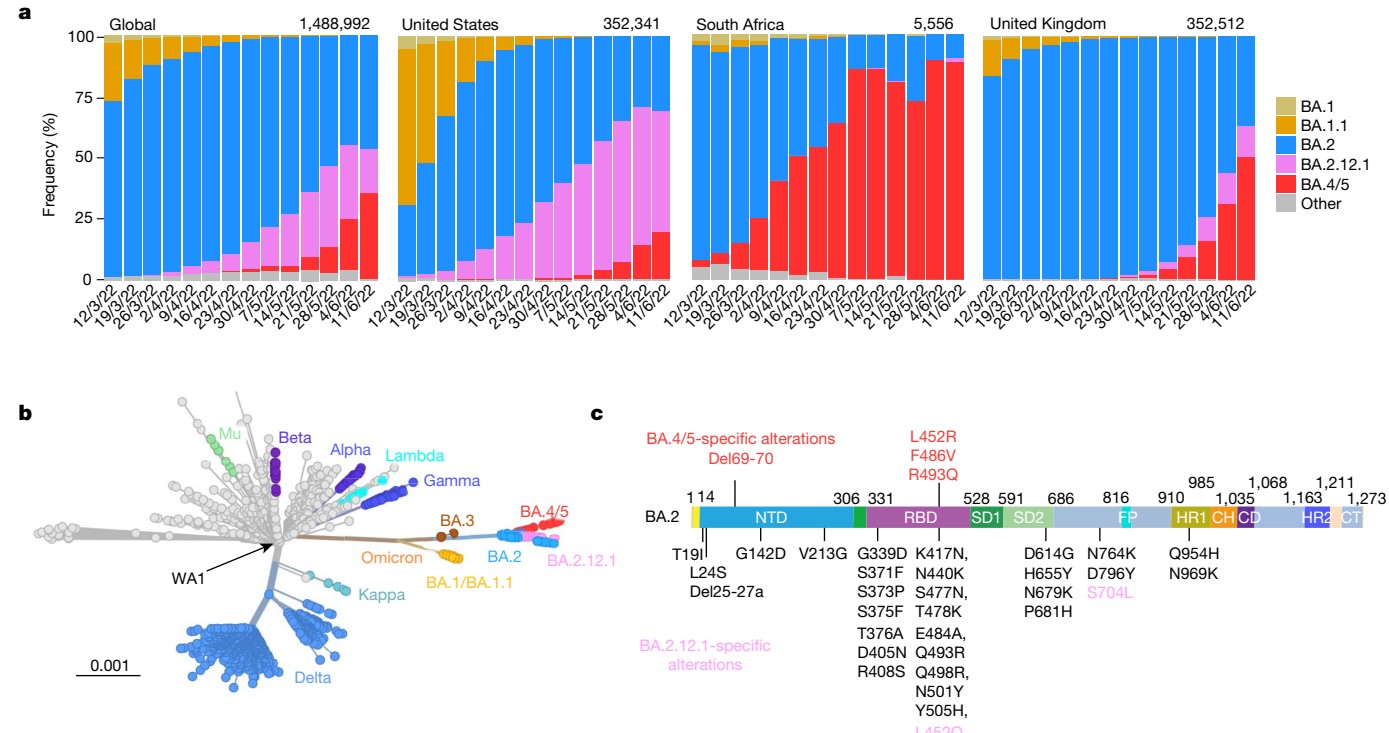

**Fig. 1 | Prevalence of SARS-CoV-2 Omicron subvariants. a**, Frequencies of BA.1, BA.1.1, BA.2, BA.2.12.1 and BA.4/5 deposited in GISAID. The value in the upper right corner of each box denotes the cumulative number of sequences for all circulating viruses in the denoted time period. **b**, Unrooted phylogenetic tree of Omicron and its subvariants along with other main SARS-CoV-2 variants. The scale bar indicates the genetic distance. **c**, Key spike mutations found in BA.2, BA.2.12.1, BA.4 and BA.5. Del, deletion.

Neutralization profiles were similar for BA.2 and BA.2.12.1 except for three class 3 RBD mAbs (Brii-198, REGN10987 and COV2-2130) that were either inactive or further impaired against the latter subvariant. Compared to BA.2 and BA.2.12.1, BA.4/5 showed substantially greater neutralization resistance to two class 2 RBD mAbs (ZCB11 and COV2-2196) as well as modest resistance to two class 3 RBD mAbs (REGN10987 and COV2-2130). Collectively, these differences indicate that mutations in BA.2.12.1 confer greater evasion from antibodies to class 3 region of RBD, whereas mutations in BA.4/5 confer greater evasion from antibodies to class 2 and class 3 regions. Only four RBD mAbs (CAB-A17, COV2-2130, 2–7 and LY-COV1404) retained good in vitro potency against both BA.2.12.1 and BA.4/5 with a half-maximum inhibitory concentration (IC$_{50}$) below 0.1 μg ml$^{−1}$. Among these four mAbs, COV2-2130 (cilgavimab) is one component of a combination known as Evusheld that is authorized for prevention of COVID-19, whereas only LY-COV1404 or bebtelovimab is authorized for therapeutic use in the clinic. For antibody combinations previously authorized or approved for clinical use, all showed a substantial loss of activity in vitro against BA.2.12.1 and BA.4/5. As for a mAb directed to the antigenic supersite of the NTD[19], 4–18 lost neutralizing activity against all Omicron subvariants. A mAb to the NTD alternate site, 5–7 (ref. [20]), was also inactive against BA.2.12.1 and BA.4/5 but retained modest activity against BA.1 and BA.1.1 (Fig. 2a).

A subset of the pseudovirus neutralization data was confirmed for four monoclonal antibodies (COV2-2196, ZCB11, REGN10987 and LY-CoV1404) in neutralization experiments using authentic viruses BA.2 and BA.4 (Extended Data Fig. 1b and Extended Data Table 1). Similar neutralization patterns were observed in the two assays, although the precise 50% neutralizing titres were different.

To identify the mutations in BA.2.12.1 and BA.4/5 that confer antibody resistance, we assessed the neutralization sensitivity of pseudoviruses carrying each of the point mutations in the background of D614G or BA.2 to the aforementioned panel of mAbs and combinations. Detailed findings are presented in Extended Data Figs. 2 and 3 and Extended Data

Table 2, and the most salient results are highlighted in Fig. 2b and discussed here. Substitutions (M, R and Q) at residue L452, previously found in the Delta and Lambda variants[21,22], conferred resistance largely to classes 2 and 3 RBD mAbs, with L452R being the more detrimental mutation. F486V broadly impaired the neutralizing activity of several class 1 and 2 RBD mAbs. Notably, this mutation decreased the potency of ZCB11 2,000-fold. By contrast, the reversion mutation R493Q sensitized BA.2 to neutralization by several class 1 and 2 RBD mAbs. This finding is consistent with our previous study[23] showing that Q493R found in the earlier Omicron subvariants mediated resistance to the same set of mAbs. L452, F486 and Q493, situated at the top of RBD, are among the residues most commonly targeted by SARS-CoV-2 neutralizing mAbs whose epitopes have been defined (Fig. 2c). In silico structural analysis showed that both L452R and L452Q caused steric hindrance to the binding by class 2 RBD mAbs. One such example is shown for LY-CoV555 (Fig. 2d), demonstrating the greater clash because of the arginine substitution and explaining why this particular mutation led to a larger loss of virus-neutralizing activity (Fig. 2b). Structural modelling of the F486V again showed steric hindrance to binding by class 2 RBD mAbs such as REGN10933, LY-CoV555 and 2–15 caused by the valine substitution (Fig. 2e).

## Receptor affinity

Epidemiological data clearly indicate that both BA.2.12.1 and BA.4/5 are very transmissible (Fig. 1a); however, the further mutations at the top of RBD (Fig. 2c) of these subvariants raises the possibility of a significant loss of affinity for the viral receptor, human angiotensin-converting enzyme 2 (hACE2). We therefore measured the binding affinity of purified spike proteins of D614G and main Omicron subvariants to dimeric hACE2 using surface plasmon resonance (SPR). The spike proteins of the Omicron subvariants exhibited similar binding affinities to hACE2, with $K_D$ values ranging from 1.66 nM for BA.4/5 to 2.36 nM for BA.2.12.1 to 2.79 nM for BA.1.1 (Fig. 3a). Despite having ≥17 mutations in the RBD

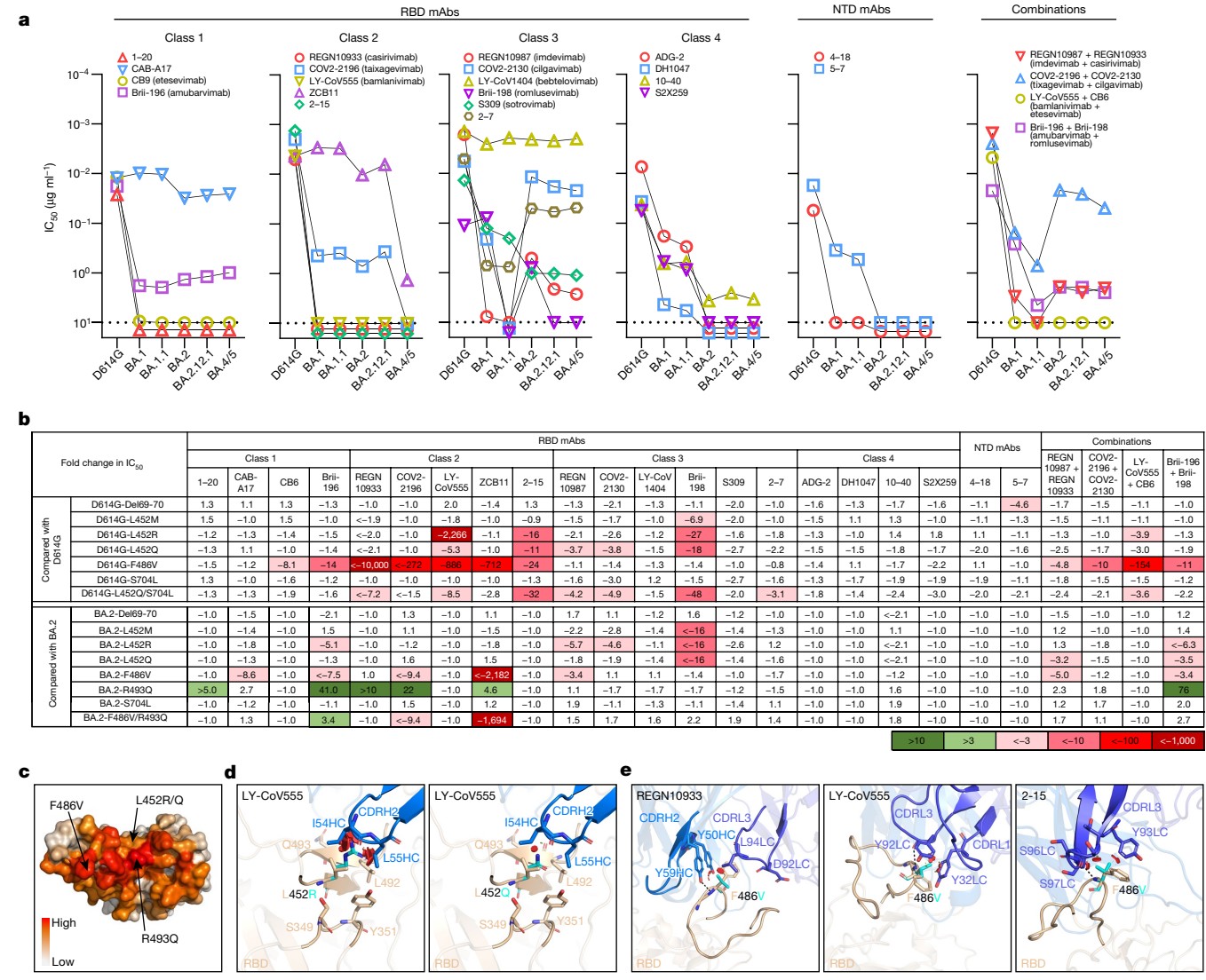

**Fig. 2 | Resistance of Omicron subvariants to neutralization by monoclonal antibodies. a**, Neutralization of D614G and Omicron subvariants by RBD- and NTD-directed mAbs. Values above the limit of detection of 10 μg ml⁻¹ (dotted line) are arbitrarily plotted to allow for visualization of each sample. **b**, Fold change in $IC_{50}$ values of point mutants relative to D614G or BA.2, with resistance coloured red and sensitization coloured green. **c**, Location of F486V, L452R/Q and R493Q on D614G RBD, with the colour indicating the per residue frequency recognized by SARS-CoV-2 neutralizing antibodies. **d**,**e**, Modelling of L452R/Q (**d**) and F486V (**e**) affect class 2 mAb neutralization. The clashes are shown with red plates; the hydrogen bonds are shown with dark dashed lines. The results shown in **a** and **b** are representative of those obtained in two independent experiments.

including some that mediate antibody escape, BA.2.12.1 and BA.4/5 also evolved concurrently to gain a slightly higher affinity for the receptor than an ancestral SARS-CoV-2, D614G ($K_D$ 5.20 nM).

To support the findings by SPR and to probe the role of point mutations in hACE2 binding, we tested BA.2, BA.2.12.1, and BA.4/5 pseudoviruses, as well as pseudoviruses containing key mutations, for their neutralization by dimeric hACE2 in vitro. The 50% inhibitory concentration ($IC_{50}$) values were lower for BA.4/5 and BA.2.12.1 than that for BA.2 (Fig. 3b), again indicating that these two emerging Omicron subvariants have not lost receptor affinity. Our results also showed that the F486V mutation compromised receptor affinity, as previously reported[24], while the R493Q reversion mutation improved receptor affinity. To structurally interpret these results, we modelled F486V and R493Q mutations on the basis of the crystal structure of BA.1-RBD–hACE2 complex[25] overlaid with ligand-free BA.2 RBD (Protein Data Bank (PDB) 7U0N and 7UB0). This analysis found that both R493 and F486 are

conformationally similar between BA.1 and BA.2, and F486V led to a loss of interaction with a hydrophobic pocket in hACE2 (Fig. 3c). On the other hand, the R493Q reversion mutation restored a hydrogen bond with H34 and avoided the charge repulsion by K31, seemingly having the opposite effect of F486V. The mutation frequency at F486 had been exceedingly low (<10 × 10⁻⁵) until the emergence of BA.4/5 (Extended Data Table 3), probably because of a compromised receptor affinity. Taken together, our findings in Figs. 2 and 3 suggest that F486V allowed BA.4 and BA.5 to extend antibody evasion while R493Q compensated to regain fitness in receptor binding.

## Neutralization by polyclonal sera

We next assessed the extent of BA.2.12.1 and BA.4/5 resistance to neutralization by sera from four different clinical cohorts. Sera from people immunized with only two doses of COVID-19 messenger RNA vaccines

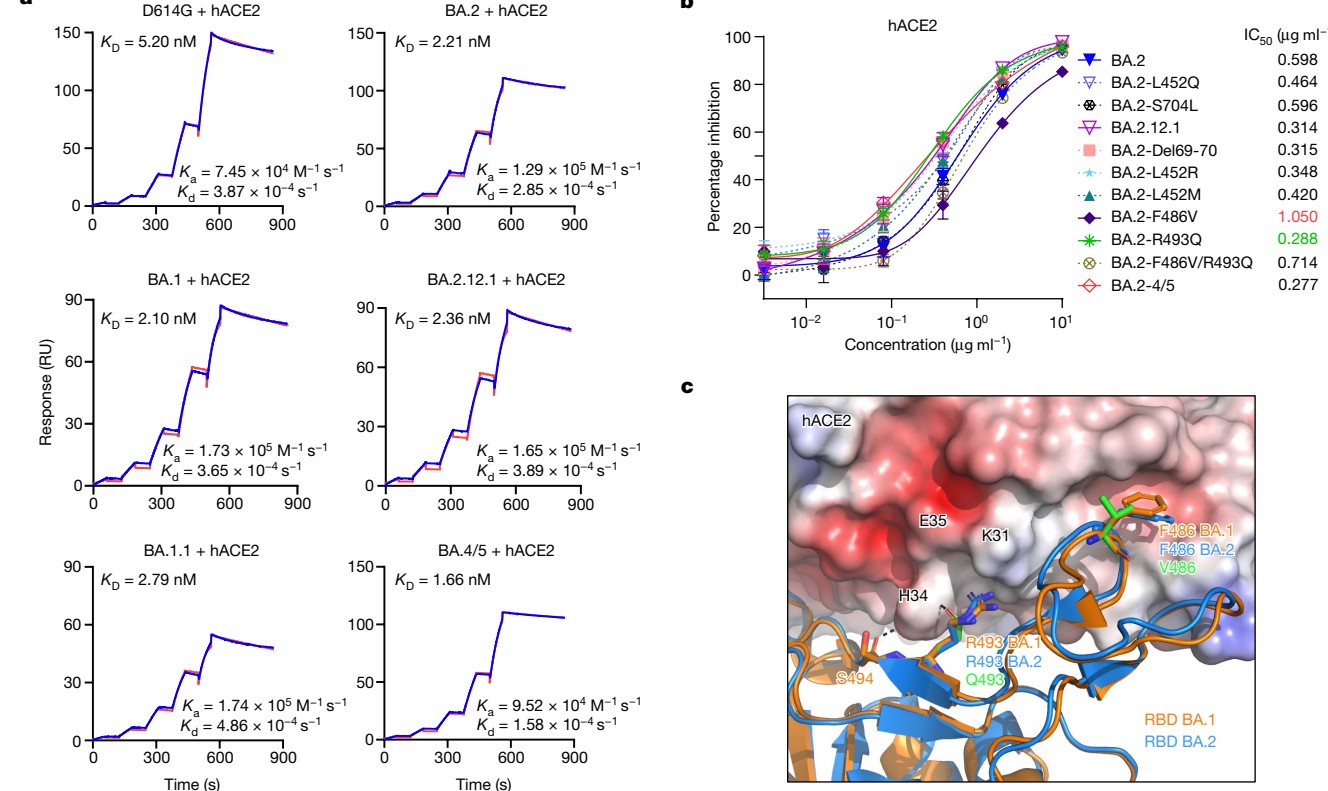

**Fig. 3 | Affinity of the spike proteins of SARS-CoV-2 Omicron subvariants to hACE2. a**, Binding affinities of Omicron subvariant S2P spike proteins to hACE2 as measured by SPR. **b**, Sensitivity of pseudotyped Omicron subvariants and the individual mutations in the background of BA.2 to hACE2 inhibition. The hACE2 concentrations resulting in 50% inhibition of infectivity ($IC_{50}$) are presented. Data are shown as mean ± standard error of mean (s.e.m.) for three technical replicates. **c**, In silico analysis for how R493Q and F486V affect hACE2 binding. The hACE2 surface is shown with charge potential, with red and blue representing negative and positive charges, respectively. Omicron BA.1 RBD in complex with hACE2 was downloaded from PDB 7U0N, and the ligand-free BA.2 RBD was downloaded from PDB 7UB0. The results shown in **a** and **b** are representative of those obtained in two independent experiments.

were not examined because most of them could not neutralize earlier Omicron subvariants[23,26]. Instead, we measured serum neutralizing activity for people who received three shots of mRNA vaccines (boosted), individuals who received mRNA vaccines before or after non-Omicron infection and patients with either BA.1 or BA.2 breakthrough infection after vaccination. Their clinical information is described in Extended Data Table 4, and the serum neutralization profiles are presented in Extended Data Fig. 4 and the 50% inhibitory dose ($ID_{50}$) titres are summarized in Fig. 4a. For the 'boosted' cohort, neutralization titres were noticeably lower (4.6- to 6.2-fold) for BA.1, BA.1.1 and BA.2 compared to D614G (Fig. 4b), as previously reported[23,26]. Titres for BA.2.12.1 and BA.4/5 were even lower, by 8.1- and 19.2-fold, respectively, relative to D614G and by 1.8- and 4.2-fold, respectively, relative to BA.2. A similar trend was observed for serum neutralization for the other cohorts, with the lowest titre against BA.4/5, followed next by titre against BA.2.12.1. Relative to BA.2, BA.2.12.1 and BA.4/5 showed 1.2–1.4-fold and 1.6–4.3-fold, respectively, greater resistance to neutralization by sera from these individuals who had both the mRNA vaccination and SARS-CoV-2 infection. In addition, sera from vaccinated and boosted individuals were assayed for neutralization of authentic viruses (Extended Data Fig. 4e,f). Neutralization titres for BA.4 were 2.7-fold lower on average compared to titres for BA.2, in line with the pseudovirus results.

We also conducted serum neutralization assays on pseudoviruses containing point mutations found in BA.2.12.1 or BA.4/5 in the background of BA.2. Del69-70, L452M/R/Q and F486V each modestly (1.1- to 2.4-fold) decreased the neutralizing activity of sera from all cohorts, while the R493Q reversion mutation modestly (roughly 1.5-fold) enhanced the neutralization (Fig. 4c and Extended Data Fig. 5). S704L, a mutation close to the S1/S2 cleavage site, did not appreciably alter

the serum neutralization titres against BA.2. For boosted serum samples, the impact of each point mutant on neutralization resistance was quantified and summarized in Fig. 4b.

Using these serum neutralization results, we then constructed a graphic display to map antigenic distances among D614G, various Omicron subvariants, and individual point mutants using only results from the boosted serum samples to avoid confounding effects from differences in clinical histories in the other cohorts. Using methods well established in influenza research[27], all virus and serum positions on the antigenic map were optimized so that the distances between them correspond to the fold drop in neutralizing $ID_{50}$ titre relative to the maximum titre for each serum. Each unit of distance in any direction on the antigenic map corresponds to a two-fold change in $ID_{50}$ titre. The resultant antigenic cartography (Fig. 4d) shows that BA.1, BA.1.1 and BA.2 are roughly equidistant from the boosted sera, with each about 2–3 antigenic units away. BA.2.12.1 is further away from BA.2 by about 1 antigenic unit. Most of all, BA.4/5 is 4.3 antigenic units further from boosted sera than D614G, and 2 antigenic units further than BA.2. Each of the point mutants Del69-70, L452M/Q/R and F486V adds antigenic distance from these sera compared to BA.2 and D614G, whereas R493Q has the opposite effect. Overall, this map makes clear that BA.4/5 is substantially more neutralization resistant to sera obtained from boosted individuals, with several mutations contributing to the antibody evasion.

## Discussion

We have systematically evaluated the antigenic properties of SARS-CoV-2 Omicron subvariants BA.2.12.1 and BA.4/5, which are rapidly expanding globally (Fig. 1a). It is apparent that BA.2.12.1 is only

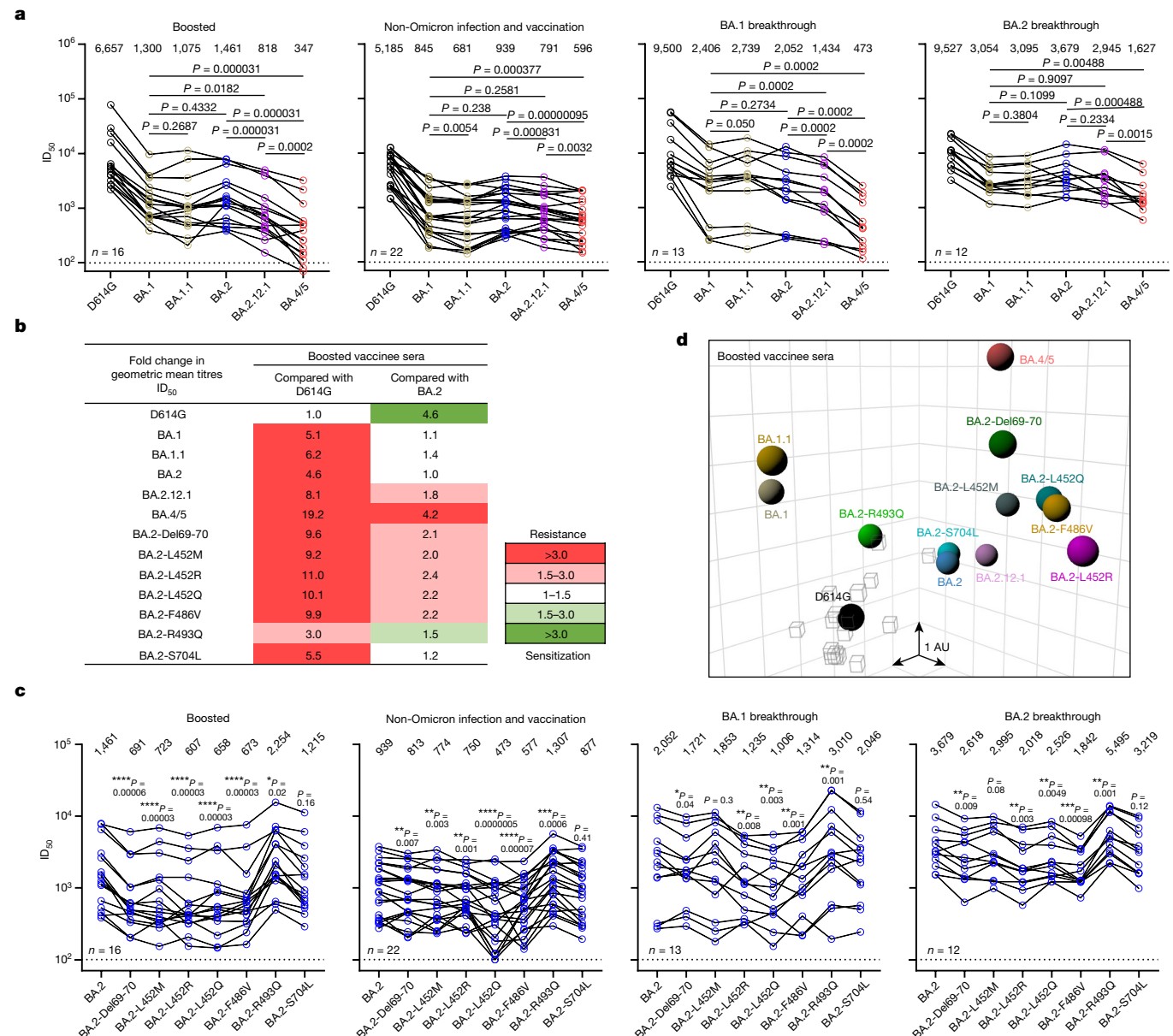

**Fig. 4 | BA.2.12.1 and BA.4/5 exhibit greater serum neutralization resistance profiles relative to BA.2. a**, Neutralization of pseudotyped D614G and Omicron subvariants by sera from four different clinical cohorts. **b**, Fold change in geometric mean $ID_{50}$ titres of boosted vaccinee sera relative to D614G and BA.2, with resistance coloured red and sensitization coloured green. **c**, Serum neutralization of BA.2 pseudoviruses containing single mutations found within BA.2.12.1 and BA.4/5. **d**, Antigenic map based on the neutralization data of boosted vaccinee sera. SARS-CoV-2 variants are shown as coloured circles and sera are shown as grey squares. The *x*, *y* and *z* axis represent antigenic units (AU) with one grid corresponding to a two-old serum dilution of the neutralization titre. An interactive map is available online (https://figshare.com/articles/media/OmicronAntigenicMap/19854046). The map orientation in the *x*, *y* and *z* axis is free to rotate. For all the panels in **a** and **c**, values above the symbols denote the geometric mean $ID_{50}$ values and values on the lower left show the sample size (*n*) for each group. *P* values were determined by using two-tailed Wilcoxon matched-pairs signed-rank tests. The results shown are representative of those obtained in two independent experiments.

modestly (1.8-fold) more resistant to sera from vaccinated and boosted individuals than the BA.2 subvariant that currently dominates the global pandemic (Fig. 4b). On the other hand, BA.4/5 is substantially (4.2-fold) more resistant, a finding consistent with results recently posted by other groups[1,28]. This antigenic distance is similar to that between the Delta variant and the ancestral virus[29] and thus is likely to lead to more breakthrough infections in the coming months. A key question now is whether BA.4/5 would out-compete BA.2.12.1, which poses less of an antigenic threat. This competition is now playing out in the United Kingdom. These new Omicron subvariants were first detected there almost simultaneously in late March of 2022. However, BA.2.12.1 now

accounts for 13% of new infections in the United Kingdom, whereas the frequency is over 50% for BA.4/5 (Fig. 1a), suggesting a transmission advantage for the latter.

Epidemiologically, as both of these two Omicron subvariants have a clear advantage in transmission, it is therefore not surprising that their abilities to bind the hACE2 receptor remain robust (Fig. 3a) despite numerous mutations in the spike protein. In fact, BA.4/5 may have slightly higher affinity for the receptor, consistent with suggestions that it might be more fit[30]. However, assessment of transmissibility would be more revealing by conducting studies on BA.2.12.1 and BA.4/5 in animal models[31].

Our studies on the specific mutations found in BA.2.12.1 and BA.4/5 show that Del69-70, L452M/R/Q and F486V could individually contribute to antibody resistance, whereas R493Q confers antibody sensitivity (Fig. 4b). Moreover, the data generated using SARS-CoV-2-neutralizing mAbs indicate that a mutation at L452 allows escape from class 2 and class 3 RBD antibodies and that the F486V mutation mediates escape from class 1 and class 2 RBD antibodies (Fig. 2b). It is not clear how Del69-70, a mutation that might increase infectivity[32] and previously seen in the Alpha variant[33], contributes to antibody resistance except for the possible evasion from certain neutralizing antibodies directed to the NTD. As for the use of clinically authorized mAbs to treat or block infection by BA.2.12.1 or BA.4/5, only bebtelovimab (LY-COV1404)[11] retains potency, whereas the combination of tixagevimab and cilgavimab (COV2-2196 and COV2-2130)[6] shows a modest loss of activity (Fig. 2a).

As the Omicron lineage has evolved over the past few months (Fig. 1b), each successive subvariant has seemingly become better and better at human transmission (Fig. 1a) as well as in antibody evasion[23,34]. It is only natural that scientific attention remains intently focused on each new subvariant of Omicron. However, we must be mindful that each of the globally dominant variants of SARS-CoV-2 (Alpha, Delta and Omicron) emerged stochastically and unexpectedly. Vigilance in our collective surveillance effort must be sustained.

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

# Methods

## Data reporting

No statistical methods were used to predetermine sample size. The experiments were not randomized and the investigators were not blinded to allocation during experiments and outcome assessment.

## Serum samples

Sera from individuals who received three doses of the mRNA-1273 or BNT162b2 vaccine were collected at Columbia University Irving Medical Center. Sera from individuals who were infected by non-Omicron variants of SARS-CoV-2 in addition to vaccination were collected from January 2021 to September 2021 at Columbia University Irving Medical Center or at the Hackensack Meridian Center for Discovery and Innovation. Sera from individuals who were infected by Omicron (BA.1 or BA.2) following vaccinations were collected from December 2021 to May 2022 at Columbia University Irving Medical Center. All samples were confirmed for previous SARS-CoV-2 infection status by antinucleoprotein ELISA. All collections were conducted under protocols reviewed and approved by the Institutional Review Board of Columbia University or the Hackensack Meridian Center for Discovery and Innovation. All participants provided written informed consent. Clinical information on the different cohorts of study participants is provided in Extended Data Table 4.

## Monoclonal antibodies

Antibodies were expressed as previously described[17]. Heavy chain variable and light chain variable genes for each antibody were synthesized (GenScript), then transfected into Expi293 cells (Thermo Fisher Scientific) and purified from the supernatant by affinity purification using rProtein A Sepharose (GE). REGN10987, REGN10933, COV2-2196 and COV2-2130 were provided by Regeneron Pharmaceuticals; Brii-196 and Brii-198 were provided by Brii Biosciences; CB6 was provided by B. Zhang and P. Kwong (National Institutes of Health (NIH)) and ZCB11 was provided by Z. Chen (University of Hong Kong).

## Cell lines

Expi293 cells were obtained from Thermo Fisher Scientific (A14527); Vero-E6 cells were obtained from the ATCC (CRL-1586); human embryonic kidney 293T (HEK293T) cells were obtained from the ATCC (CRL-3216). Cells were purchased from authenticated vendors and morphology was confirmed visually before use. All cell lines tested mycoplasma negative.

## Variant SARS-CoV-2 spike plasmid construction

BA.1, BA.1.1 and BA.2 spike-expressing plasmids were generated as previously described[23,26]. Plasmids encoding the BA.2.12.1 and BA.4/5 spikes, as well as the individual and double mutations found in BA.2.12.1 and BA.4/5, were generated using the QuikChange II XL site-directed mutagenesis kit according to the manufacturer's instructions (Agilent). To make the constructs for expression of stabilized soluble S2P spike trimer proteins, 2P substitutions (K986P and V987P) and a 'GSAS' substitution of the furin cleavage site (682–685 amino acids (aa) in WA1) were introduced into the spike-expressing plasmids[35], and then the ectodomain (1–1,208 aa in WA1) of the spike was fused with a C-terminal 8× His-tag and cloned into the paH vector. All constructs were confirmed by Sanger sequencing.

## Expression and purification of SARS-CoV-2 S2P spike proteins

SARS-CoV-2 S2P spike trimer proteins of the D614G and Omicron subvariants were generated by transfecting Expi293 cells with the S2P spike trimer-expressing constructs using 1 mg ml$^{-1}$ polyethylenimine and then purifying from the supernatants 5 days post-transfection using Ni-NTA resin (Invitrogen) according to the manufacturer's instructions[17].

## SPR

SPR binding assays for hACE2 binding to SARS-CoV-2 S2P spike were performed using a Biacore T200 biosensor equipped with a Series S CM5 chip (Cytiva), in a running buffer of 10 mM HEPES pH 7.4, 150 mM NaCl, 3 mM EDTA, 0.05% P-20 (Cytiva) at 25 °C. Spike proteins were captured through their C-terminal His-tag over an anti-His antibody surface. These surfaces were generated using the His-capture kit (Cytiva) according to the manufacturer's instructions, resulting in roughly 10,000 RU of anti-His antibody over each surface. An anti-His antibody surface without antigen was used as a reference flow cell to remove bulk shift changes from the binding signal.

Binding of human ACE2-Fc protein (Sino Biological) was tested using a threefold dilution series with concentrations ranging from 2.46 to 200 nM. The association and dissociation rates were each monitored for 60 and 300 s, respectively, at 30 μl min$^{-1}$. The bound spike–ACE2 complex was regenerated from the anti-His antibody surface using 10 mM glycine pH 1.5. Blank buffer cycles were performed by injecting running buffer instead of human ACE2-Fc to remove systematic noise from the binding signal. The resulting data were processed and fit to a 1:1 binding model using Biacore Evaluation Software.

## Pseudovirus production

Pseudoviruses were produced in the vesicular stomatitis virus background, in which the native glycoprotein was replaced by that of SARS-CoV-2 and its variants, as previously described[17]. In brief, HEK293T cells were transfected with a spike expression construct with 1 mg ml$^{-1}$ polyethylenimine and cultured overnight at 37 °C under 5% $CO_2$, and then infected with vesicular stomatitis virus-G pseudotyped ΔG-luciferase (G*ΔG-luciferase, Kerafast) 1 day post-transfection. After 2 h of infection, cells were washed three times, changed to fresh medium and then cultured for around another 24 h before the supernatants were collected, clarified by centrifugation and aliquoted and stored at −80 °C for further use.

## Pseudovirus neutralization assay

All viruses were first titrated to normalize the viral input between assays. Heat-inactivated sera or antibodies were first serially diluted (fivefold) in medium in 96-well plates in triplicate, starting at 1:100 dilution for sera and 10 μg ml$^{-1}$ for antibodies. Pseudoviruses were then added and the virus–sample mixture was incubated at 37 °C for 1 h. Vero-E6 cells were then added at a density of $3 \times 10^4$ cells per well and the plates were incubated at 37 °C for about 10 h. Luciferase activity was quantified using the Luciferase Assay System (Promega) according to the manufacturer's instructions using SoftMax Pro v.7.0.2 (Molecular Devices). Neutralization curves and IC$_{50}$ values were derived by fitting a non-linear five-parameter dose-response curve to the data in GraphPad Prism v.9.2.

## Authentic virus neutralization assay

The SARS-CoV-2 viruses hCoV-19/USA/CO-CDPHE-2102544747/2021 (BA.2) and hCoV-19/USA/MD-HP30386/2022 (BA.4) were obtained from BEI Resources (NIAID, NIH) and propagated by passaging in Vero-E6 cells. Virus infectious titres were determined by an end-point dilution and cytopathogenic effect assay on Vero-E6 cells as previously described[17].

An end-point dilution microplate neutralization assay was performed to measure the neutralization activity of sera from vaccinated and boosted individuals as well as of purified monoclonal antibodies. In brief, serum samples were subjected to successive fivefold dilutions starting from 1:100. Monoclonal antibodies were serially diluted (fivefold) starting at 5 μg ml$^{-1}$. Triplicates of each dilution were incubated with SARS-CoV-2 at a multiplicity of infection of 0.1 in Eagle's minimum essential medium with 7.5% inactivated foetal calf serum for 1 h at 37 °C. After incubation, the virus–antibody mixture was transferred onto a monolayer of Vero-E6 cells grown overnight. The cells were

incubated with the mixture for around 70 h. Cytopathogenic effects of viral infection were visually scored for each well in a blinded manner by two independent observers. The results were then converted into the percentage of neutralization at a given sample dilution or monoclonal antibody concentration, and the data (mean ± s.e.m.) were plotted using a five-parameter dose-response curve in GraphPad Prism v.9.2.

**Antibody targeting frequency and mutagenesis analysis for RBD**
The SARS-CoV-2 spike structure (PDB 6ZGE) used for showing epitope footprints was downloaded from the PDB. Epitope residues were identified using PISA[36] with default parameters, and the RBD residues with non-zero buried accessible surface area were considered epitope residues. For each residue within the RBD, the frequency of antibody recognition was calculated as the number of contact antibodies[37]. The structures of antibody–spike complexes for modelling were also obtained from PDB (7L5B (2–15), 6XDG (REGN10933) and 7KMG (LY-CoV555)). Omicron BA.1 RBD in complex with hACE2 was downloaded from PDB 7U0N, and the ligand-free BA.2 RBD was downloaded from PDB 7UB0. PyMOL v.2.3.2 was used to perform mutagenesis and to generate structural plots (Schrödinger, LLC).

**Antigenic cartography**
The antigenic distances between SARS-CoV-2 variants were approximated by incorporating the neutralization potency of each serum sample into a previously described antigenic cartography approach[27]. The map was generated by the Racmacs package (https://acorg.github.io/Racmacs/, v.1.1.4) in R with the optimization steps set to 2,000 and with the minimum column basis parameter set to 'none'.

**Reporting summary**
Further information on research design is available in the Nature Research Reporting Summary linked to this article.

## Data availability
All data are provided in the paper. Materials in this study will be made available under an appropriate Materials Transfer Agreement.

Sequences for Omicron prevalence analysis were downloaded from the global initiative on sharing all influenza data (GISAID) (https://www.gisaid.org/). The structures used for analysis in this study are available from PDB under IDs 6ZGE, 7L5B, 6XDG, 7U0N, 7UB0 and 7KMG. The interactive antigenic map based on the neutralization data of boosted vaccine sera in Fig. 4d is available online (https://figshare.com/articles/media/OmicronAntigenicMap/19854046).

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

**Acknowledgements** This study was supported by funding from the Gates Foundation, JPB Foundation, A. and P. Cherng, S. Yin, C. Ludwig, D. and R. Wu, Regeneron Pharmaceuticals and the NIH SARS-CoV-2 Assessment of Viral Evolution (SAVE) Program. We acknowledge D.S. Perlin for providing serum samples from a few patients with COVID-19. We thank all who contributed their data to GISIAD.

**Author contributions** D.D.H. and L.L. conceived this project. Q.W. and L.L. conducted pseudovirus neutralization experiments and purified SARS-CoV-2 spike proteins. Y.G. and Z.S. conducted bioinformatic analyses. Q.W., L.L. and S.I. constructed the spike expression plasmids. Q.W. managed the project. J.Y., M.W. and Z.C. expressed and purified antibodies. L.L. and Z.L. performed the SPR assay. M.T.Y., M.E.S., J.Y.C., A.D.B. J.G.S., N.N. and K.M. provided clinical samples. H.M. aided sample collections. M.S.N. and Y.H. performed infectious SARS-CoV-2 neutralization assays. D.D.H. and L.L. directed and supervised the project. Q.W., Y.G., L.L. and D.D.H. analysed the results and wrote the manuscript.

**Competing interests** S.I, J.Y., Y.H., L.L. and D.D.H. are inventors on patent applications (WO2021236998) or provisional patent applications (63/271,627) filed by Columbia University for several SARS-CoV-2 neutralizing antibodies described in this paper. Both sets of applications are under review. D.D.H. is a cofounder of TaiMed Biologics and RenBio, consultant to WuXi Biologics and Brii Biosciences and board director for Vicarious Surgical.

**Additional information**
**Correspondence and requests for materials** should be addressed to Lihong Liu or David D. Ho.

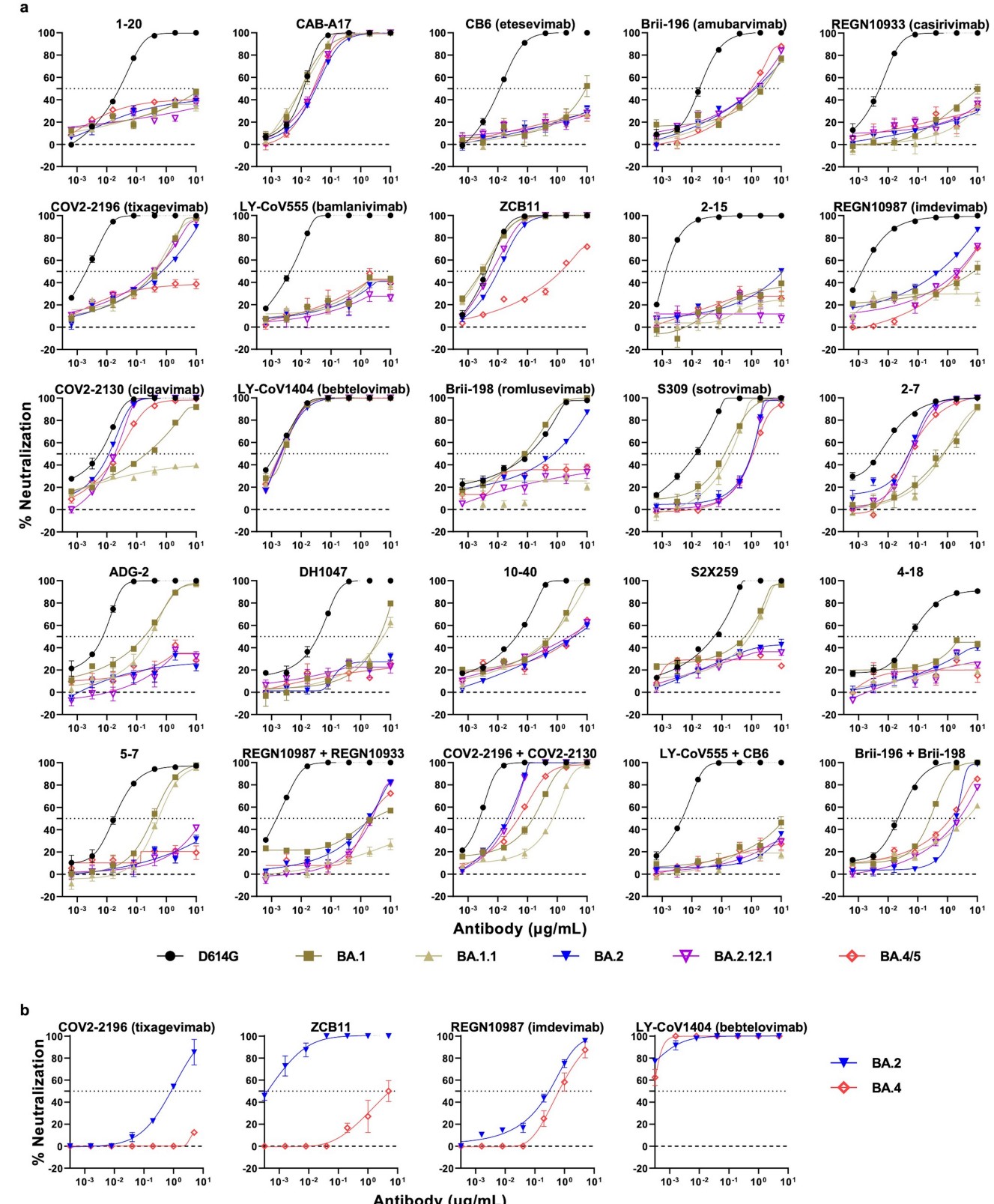

**Extended Data Fig. 1 | Pseudovirus (a) and authentic virus (b) neutralization curves of D614G and Omicron subvariants by monoclonal antibodies.** Data are shown as mean ± SEM from three technical replicates and representative of those obtained in two independent experiments.

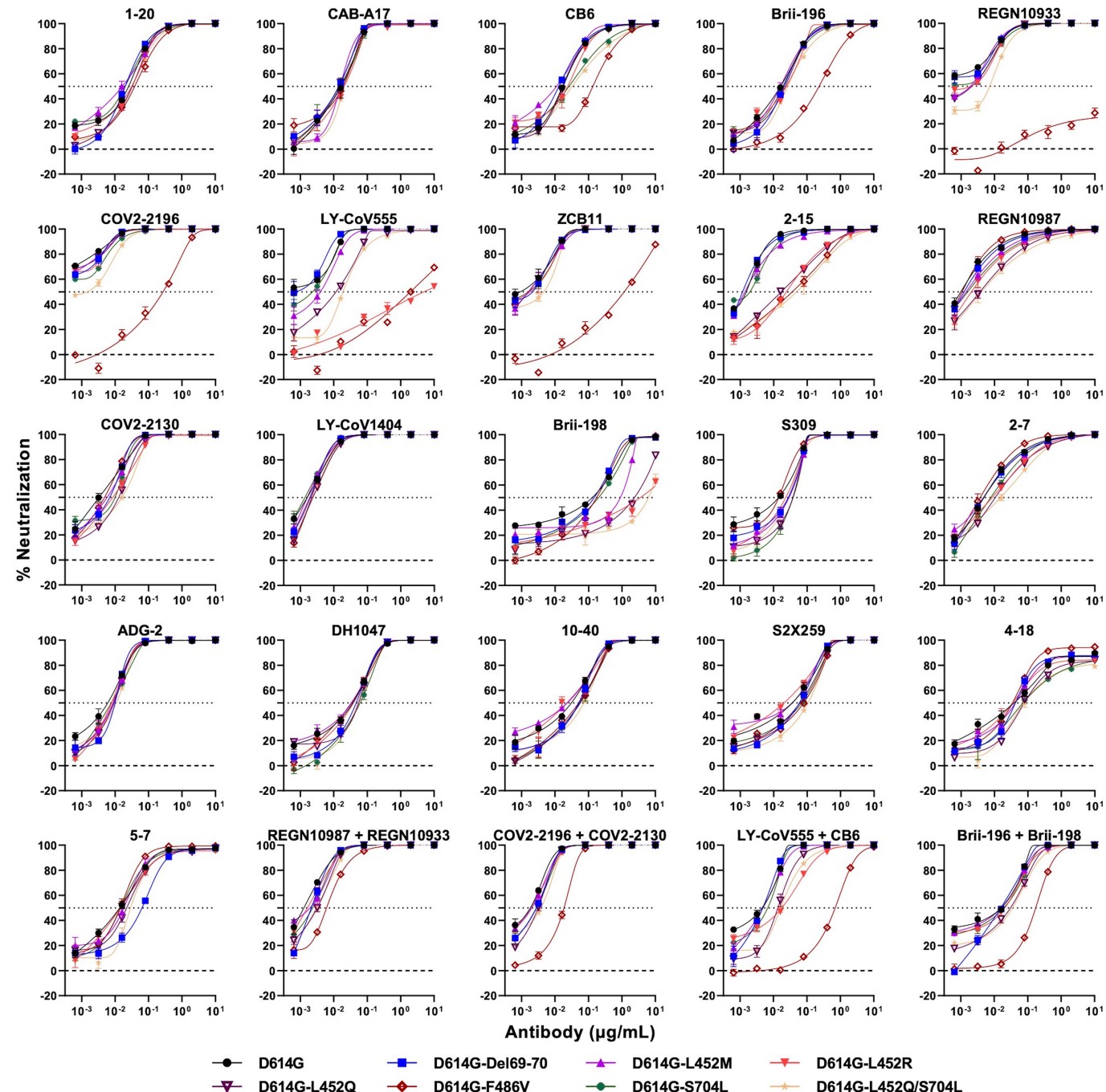

**Extended Data Fig. 2 | Pseudovirus neutralization curves for monoclonal antibodies against individual SARS-CoV-2 mutations in the background of D614G.** Data are shown as mean ± SEM from three technical replicates and representative of those obtained in two independent experiments.

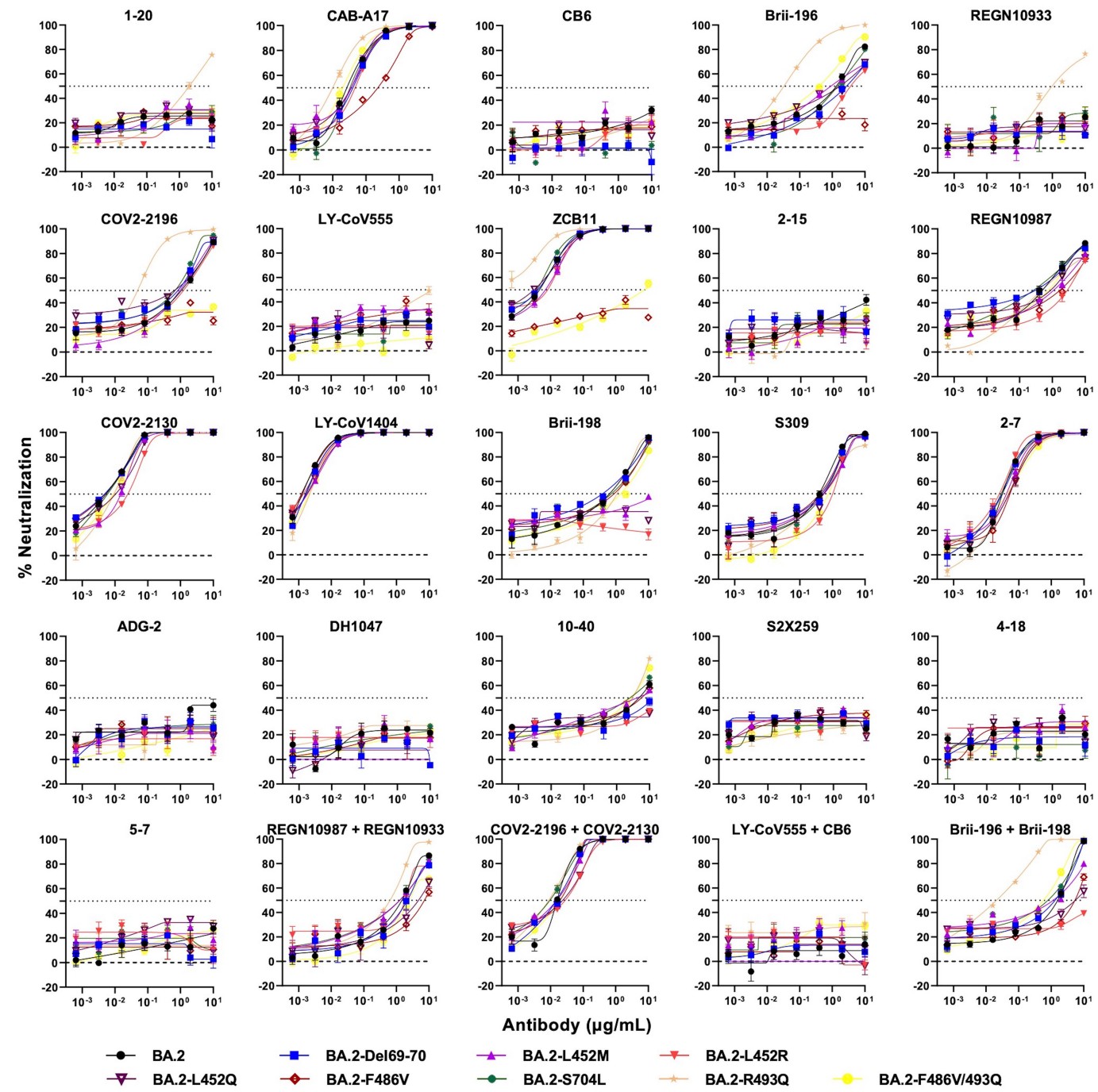

**Extended Data Fig. 3 | Pseudovirus neutralization curves for monoclonal antibodies against individual SARS-CoV-2 mutations in the background of BA.2.** Data are shown as mean ± SEM from three technical replicates and representative of those obtained in two independent experiments.

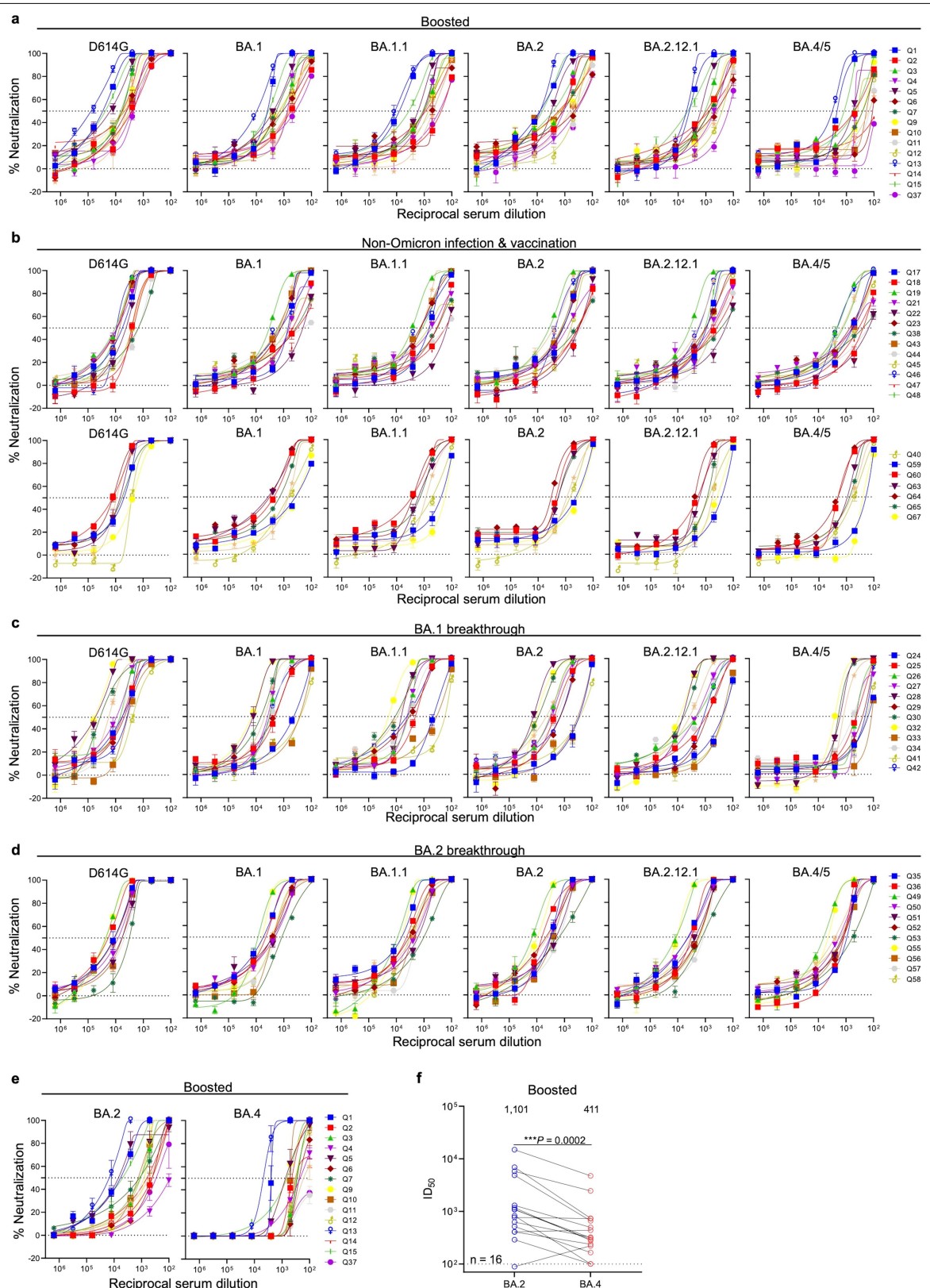

**Extended Data Fig. 4 | Neutralization curves of serum against D614G and Omicron subvariants.** Neutralization by **a**, boosted vaccinee sera on pseudoviruses. **b**, non-Omicron infection & vaccination sera on pseudoviruses. **c**, BA.1 breakthrough sera on pseudoviruses. **d**, BA.2 breakthrough sera on pseudoviruses. **e**, boosted vaccinee sera on authentic viruses. **f**, Neutralization ID$_{50}$ titers of authentic BA.2 and BA.4 by boosted vaccinee sera. Values above

the symbols denote the geometric mean ID$_{50}$ values and values on the lower left show the sample size (n). $P$ values were determined by using two-tailed Wilcoxon matched-pairs signed-rank tests. Error bars in **a**, **b**, **c**, **d**, and **e** denote mean ± SEM for three technical replicates. All data are representative of those obtained in two independent experiments.

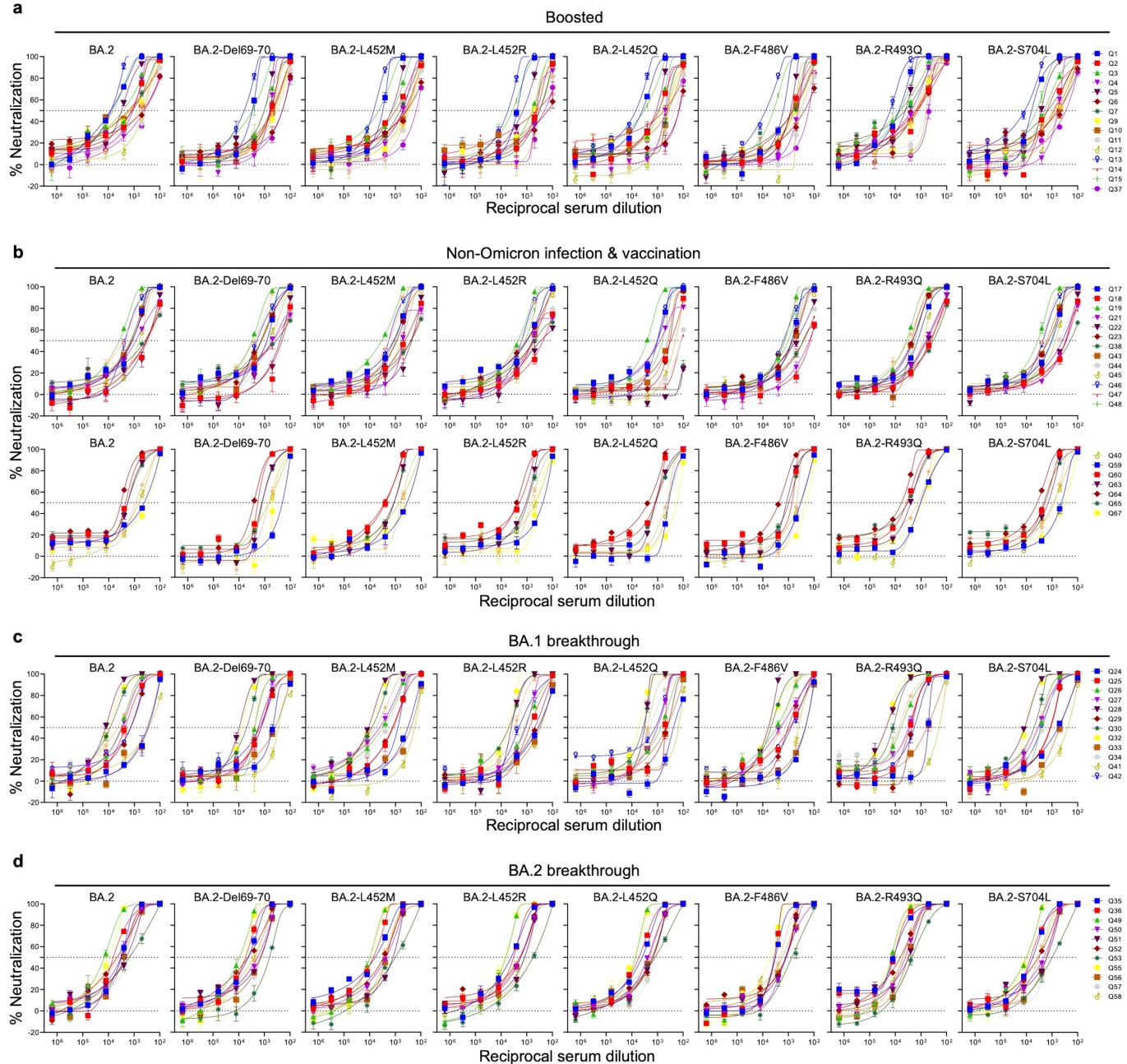

**Extended Data Fig. 5 | Pseudovirus neutralization curves of serum against BA.2 and BA.2 pseudovirus carrying individual mutations.** Neutralization by **a**, boosted vaccinee sera. **b**, non-Omicron infection & vaccination sera. **c**, BA.1 breakthrough sera. **d**, BA.2 breakthrough sera. Error bars denote mean ± SEM for three technical replicates. Data are representative of those obtained in two independent experiments.

**Extended Data Table 1 | Neutralization IC$_{50}$ values for indicated pseudoviruses (a) and authentic viruses (b) by monoclonal antibodies**

**a**

| IC$_{50}$ (µg/mL) | RBD mAbs | | | | | | | | | | | | | | | | | | | NTD mAbs | | Combination | | | |
|---|---|---|---|---|---|---|---|---|---|---|---|---|---|---|---|---|---|---|---|---|---|---|---|---|---|
| | Class 1 | | | | Class 2 | | | | | Class 3 | | | | | | Class 4 | | | | | | REGN 10987 + REGN 10933 | COV2- 2196 + COV2- 2130 | LY- CoV555 + CB6 | Brii-196 + Brii- 198 |
| | 1-20 | CAB-A17 | CB6 | Brii-196 | REGN 10933 | COV2- 2196 | LY- CoV555 | ZCB11 | 2-15 | REGN 10987 | COV2- 2130 | LY-CoV 1404 | Brii-198 | S309 | 2-7 | ADG-2 | DH1047 | 10-40 | S2X259 | 4-18 | 5-7 | | | | |
| D614G | 0.027 | 0.012 | 0.012 | 0.018 | 0.005 | 0.002 | 0.004 | 0.004 | 0.001 | 0.001 | 0.006 | 0.001 | 0.110 | 0.014 | 0.005 | 0.007 | 0.037 | 0.042 | 0.055 | 0.054 | 0.017 | 0.001 | 0.002 | 0.005 | 0.022 |
| BA.1 | >10 | 0.010 | 9.253 | 2.385 | >10 | 0.432 | >10 | 0.003 | >10 | 7.586 | 0.209 | 0.003 | 0.078 | 0.127 | 0.716 | 0.181 | 4.322 | 0.644 | 0.590 | >10 | 0.347 | 2.951 | 0.154 | >10 | 0.260 |
| BA.1.1 | >10 | 0.010 | >10 | 1.792 | >10 | 0.385 | >10 | 0.003 | >10 | >10 | >10 | 0.002 | >10 | 0.200 | 0.763 | 0.295 | 5.723 | 0.600 | 0.859 | >10 | 0.534 | >10 | 0.708 | >10 | 4.394 |
| BA.2 | >10 | 0.031 | >10 | 1.346 | >10 | 0.704 | >10 | 0.010 | >10 | 0.505 | 0.012 | 0.002 | 0.782 | 1.019 | 0.050 | >10 | >10 | 3.642 | >10 | >10 | >10 | 1.882 | 0.021 | >10 | 1.907 |
| BA.2.12.1 | >10 | 0.027 | >10 | 1.171 | >10 | 0.361 | >10 | 0.006 | >10 | 2.125 | 0.018 | 0.002 | >10 | 1.035 | 0.059 | >10 | >10 | 2.519 | >10 | >10 | >10 | 2.400 | 0.026 | >10 | 1.936 |
| BA.4/5 | >10 | 0.025 | >10 | 0.978 | >10 | >10 | >10 | 1.351 | >10 | 2.682 | 0.022 | 0.002 | >10 | 1.120 | 0.049 | >10 | >10 | 3.404 | >10 | >10 | >10 | 1.998 | 0.049 | >10 | 2.445 |

| <0.01 | <0.1 | <1 | <10 | >10 |
|---|---|---|---|---|

**b**

| IC$_{50}$ (µg/mL) | RBD mAbs | | | |
|---|---|---|---|---|
| | Class 2 | | Class 3 | |
| | COV2- 2196 | ZCB11 | REGN 10987 | LY- CoV1404 |
| BA.2 | 0.8287 | 0.0004 | 0.3057 | <0.001 |
| BA.4 | >5 | >5 | 0.6418 | <0.001 |

| <0.01 | <0.1 | <1 | <5 | >5 |
|---|---|---|---|---|

**Extended Data Table 2 | Pseudovirus neutralization IC$_{50}$ values for monoclonal antibodies against D614G (a) and BA.2 (b) carrying individual mutations**

**a**

| IC$_{50}$ (µg/mL) | RBD mAbs | | | | | | | | | | | | | | | | | | | NTD mAbs | | Combination | | | |
| | Class 1 | | | | Class 2 | | | | | Class 3 | | | | | | Class 4 | | | | | | REGN10987 + REGN10933 | COV2-2196 + COV2-2130 | LY-CoV55 5 + CB6 | Brii-196 + Brii-198 |
| | 1-20 | CAB-A17 | CB6 | Brii-196 | REGN10933 | COV2-2196 | LY-CoV555 | ZCB11 | 2-15 | REGN10987 | COV2-2130 | LY-CoV1404 | Brii-198 | S309 | 2-7 | ADG-2 | DH1047 | 10-40 | S2X259 | 4-18 | 5-7 | | | | |
|---|---|---|---|---|---|---|---|---|---|---|---|---|---|---|---|---|---|---|---|---|---|---|---|---|---|
| D614G | 0.026 | 0.016 | 0.017 | 0.016 | <0.001 | <0.001 | 0.002 | 0.002 | 0.002 | 0.001 | 0.003 | 0.002 | 0.129 | 0.014 | 0.005 | 0.006 | 0.037 | 0.031 | 0.036 | 0.038 | 0.014 | 0.001 | 0.002 | 0.005 | 0.016 |
| D614G-Del69-70 | 0.020 | 0.014 | 0.013 | 0.021 | <0.001 | <0.001 | 0.001 | 0.002 | 0.001 | 0.001 | 0.007 | 0.002 | 0.148 | 0.028 | 0.005 | 0.010 | 0.047 | 0.051 | 0.059 | 0.042 | 0.064 | 0.002 | 0.003 | 0.005 | 0.016 |
| D614G-L452M | 0.017 | 0.016 | 0.011 | 0.016 | 0.002 | <0.001 | 0.004 | 0.002 | 0.001 | 0.001 | 0.006 | 0.002 | 0.892 | 0.029 | 0.005 | 0.009 | 0.034 | 0.023 | 0.036 | 0.042 | 0.018 | 0.002 | 0.002 | 0.005 | 0.016 |
| D614G-L452R | 0.032 | 0.020 | 0.024 | 0.024 | 0.002 | <0.001 | 5.018 | 0.002 | 0.024 | 0.002 | 0.009 | 0.002 | 3.526 | 0.023 | 0.009 | 0.008 | 0.039 | 0.023 | 0.020 | 0.035 | 0.014 | 0.002 | 0.002 | 0.018 | 0.021 |
| D614G-L452Q | 0.033 | 0.014 | 0.018 | 0.023 | 0.002 | <0.001 | 0.012 | 0.002 | 0.017 | 0.004 | 0.013 | 0.002 | 2.346 | 0.038 | 0.011 | 0.009 | 0.056 | 0.055 | 0.061 | 0.078 | 0.022 | 0.003 | 0.003 | 0.013 | 0.031 |
| D614G-F486V | 0.039 | 0.019 | 0.135 | 0.231 | >10 | 0.272 | 1.961 | 1.174 | 0.036 | 0.001 | 0.005 | 0.002 | 0.175 | 0.014 | 0.004 | 0.009 | 0.033 | 0.051 | 0.079 | 0.034 | 0.014 | 0.006 | 0.019 | 0.701 | 0.174 |
| D614G-S704L | 0.020 | 0.017 | 0.026 | 0.019 | <0.001 | <0.001 | 0.002 | 0.002 | 0.002 | 0.002 | 0.010 | 0.001 | 0.199 | 0.038 | 0.008 | 0.008 | 0.061 | 0.057 | 0.069 | 0.071 | 0.015 | 0.002 | 0.003 | 0.007 | 0.019 |
| D614G-L452Q/S704L | 0.033 | 0.020 | 0.032 | 0.026 | 0.007 | 0.001 | 0.019 | 0.005 | 0.049 | 0.004 | 0.016 | 0.002 | 6.166 | 0.028 | 0.015 | 0.011 | 0.052 | 0.074 | 0.106 | 0.076 | 0.029 | 0.003 | 0.004 | 0.016 | 0.035 |

**b**

| IC$_{50}$ (µg/mL) | RBD mAbs | | | | | | | | | | | | | | | | | | | NTD mAbs | | Combination | | | |
| | Class 1 | | | | Class 2 | | | | | Class 3 | | | | | | Class 4 | | | | | | REGN10987 + REGN10933 | COV2-2196 + COV2-2130 | LY-CoV55 5 + CB6 | Brii-196 + Brii-198 |
| | 1-20 | CAB-A17 | CB6 | Brii-196 | REGN10933 | COV2-2196 | LY-CoV555 | ZCB11 | 2-15 | REGN10987 | COV2-2130 | LY-CoV1404 | Brii-198 | S309 | 2-7 | ADG-2 | DH1047 | 10-40 | S2X259 | 4-18 | 5-7 | | | | |
|---|---|---|---|---|---|---|---|---|---|---|---|---|---|---|---|---|---|---|---|---|---|---|---|---|---|
| BA.2 | >10 | 0.027 | >10 | 1.329 | >10 | 1.060 | >10 | 0.005 | >10 | 0.495 | 0.005 | 0.001 | 0.642 | 0.393 | 0.032 | >10 | >10 | 4.824 | >10 | >10 | >10 | 1.475 | 0.016 | >10 | 1.592 |
| BA.2-Del69-70 | >10 | 0.040 | >10 | 2.726 | >10 | 0.835 | >10 | 0.004 | >10 | 0.298 | 0.005 | 0.002 | 0.394 | 0.469 | 0.031 | >10 | >10 | >10 | >10 | >10 | >10 | 2.178 | 0.015 | >10 | 1.320 |
| BA.2-L452M | >10 | 0.036 | >10 | 0.907 | >10 | 0.970 | >10 | 0.007 | >10 | 1.081 | 0.015 | 0.002 | >10 | 0.557 | 0.042 | >10 | >10 | 4.246 | >10 | >10 | >10 | 1.276 | 0.015 | >10 | 1.163 |
| BA.2-L452R | >10 | 0.047 | >10 | 6.815 | >10 | 1.228 | >10 | 0.008 | >10 | 2.832 | 0.025 | 0.001 | >10 | 1.022 | 0.026 | >10 | >10 | >10 | >10 | >10 | >10 | 1.864 | 0.028 | >10 | >10 |
| BA.2-L452Q | >10 | 0.036 | >10 | 1.717 | >10 | 0.655 | >10 | 0.003 | >10 | 0.872 | 0.010 | 0.002 | >10 | 0.535 | 0.051 | >10 | >10 | >10 | >10 | >10 | >10 | 4.793 | 0.024 | >10 | 5.525 |
| BA.2-F486V | >10 | 0.229 | >10 | >10 | >10 | >10 | >10 | >10 | >10 | 1.681 | 0.005 | 0.002 | 0.887 | 0.412 | 0.054 | >10 | >10 | 5.759 | >10 | >10 | >10 | 7.366 | 0.020 | >10 | 5.377 |
| BA.2-R493Q | 2.020 | 0.010 | >10 | 0.033 | 0.960 | 0.049 | >10 | <0.001 | >10 | 0.454 | 0.009 | 0.002 | 1.089 | 0.485 | 0.049 | >10 | >10 | 3.008 | >10 | >10 | >10 | 0.641 | 0.009 | >10 | 0.021 |
| BA.2-S704L | >10 | 0.033 | >10 | 1.464 | >10 | 0.686 | >10 | 0.004 | >10 | 0.262 | 0.006 | 0.002 | 0.735 | 0.539 | 0.029 | >10 | >10 | 2.537 | >10 | >10 | >10 | 1.262 | 0.010 | >10 | 0.800 |
| BA.2-F486V/R493Q | >10 | 0.020 | >10 | 0.394 | >10 | >10 | >10 | 7.766 | >10 | 0.757 | 0.009 | 0.002 | 1.414 | 0.754 | 0.044 | >10 | >10 | 2.751 | >10 | >10 | >10 | 2.498 | 0.017 | >10 | 0.586 |

| <0.01 | <0.1 | <1 | <10 | >10 |
|---|---|---|---|---|

**Extended Data Table 3 | Mutation frequencies at position F486 within different SARS-CoV-2 variants**

| Mutation | Count in BA.1 | Frequency in BA.1 | Count in BA.2 | Frequency in BA.2 | Count in other variants | Frequency in other variants |
|---|---|---|---|---|---|---|
| F486V | 23 | 2.17E-06 | 134 | 1.26E-05 | 898 | 8.48E-05 |
| Del486 | 193 | 1.82E-05 | 549 | 5.18E-05 | 760 | 7.17E-05 |
| F486L | 37 | 3.49E-06 | 10 | 9.44E-07 | 155 | 1.46E-05 |
| F486S | 61 | 5.76E-06 | 10 | 9.44E-07 | 142 | 1.34E-05 |
| F486I | 5 | 4.72E-07 | 2 | 1.89E-07 | 34 | 3.21E-06 |
| F486Y | 12 | 1.13E-06 | 2 | 1.89E-07 | 20 | 1.89E-06 |
| F486W | 8 | 7.55E-07 | 1 | 9.44E-08 | 10 | 9.44E-07 |
| F486T | 5 | 4.72E-07 | 0 | 0 | 5 | 4.72E-07 |
| F486E | 2 | 1.89E-07 | 0 | 0 | 3 | 2.83E-07 |
| F486N | 2 | 1.89E-07 | 0 | 0 | 3 | 2.83E-07 |
| F486H | 2 | 1.89E-07 | 0 | 0 | 2 | 1.89E-07 |
| F486P | 2 | 1.89E-07 | 0 | 0 | 2 | 1.89E-07 |
| F486R | 1 | 9.44E-08 | 0 | 0 | 2 | 1.89E-07 |
| F486C | 0 | 0 | 0 | 0 | 1 | 9.44E-08 |
| F486G | 1 | 9.44E-08 | 0 | 0 | 1 | 9.44E-08 |
| F486M | 0 | 0 | 0 | 0 | 1 | 9.44E-08 |
| F486Q | 0 | 0 | 1 | 9.44E-08 | 1 | 9.44E-08 |

# Extended Data Table 4 | Demographics on the clinical cohorts

| Sample ID | Vaccine type and infected strain | Days post-vaccination or *infection (after last exposure) | Documented COVID-19 | Age | Gender |
|---|---|---|---|---|---|
| **Boosted** | | | | | |
| Q1 | mRNA-1273/mRNA-1273/mRNA-1273 | 29 | No | 66 | Female |
| Q2 | BNT162b2/BNT162b2/BNT162b2 | 30 | No | 68 | Male |
| Q3 | BNT162b2/BNT162b2/BNT162b2 | 14 | No | 64 | Female |
| Q4 | BNT162b2/BNT162b2/BNT162b2 | 34 | No | 55 | Male |
| Q5 | BNT162b2/BNT162b2/BNT162b2 | 34 | No | 45 | Male |
| Q6 | BNT162b2/BNT162b2/BNT162b2 | 15 | No | 50 | Female |
| Q7 | BNT162b2/BNT162b2/BNT162b2 | 15 | No | 48 | Female |
| Q8 | BNT162b2/BNT162b2/BNT162b2 | 29 | No | 71 | Male |
| Q9 | BNT162b2/BNT162b2/BNT162b2 | 90 | No | 59 | Male |
| Q10 | BNT162b2/BNT162b2/BNT162b2 | 33 | No | 45 | Male |
| Q11 | BNT162b2/BNT162b2/BNT162b2 | 87 | No | 66 | Female |
| Q12 | BNT162b2/BNT162b2/BNT162b2 | 84 | No | 26 | Male |
| Q13 | mRNA-1273/mRNA-1273/mRNA-1273 | 23 | No | 28 | Female |
| Q14 | BNT162b2/BNT162b2/BNT162b2 | 14 | No | 78 | Male |
| Q15 | BNT162b2/BNT162b2/mRNA-1273 | 32 | No | 39 | Male |
| Q37 | BNT162b2/BNT162b2/BNT162b2 | 20 | No | Unknown | Female |
| **Non-Omicron infection & vaccination** | | | | | |
| Q17 | R.1/mRNA-1273/mRNA-1273 | 7 | Yes | 34 | Female |
| Q18 | R.1/mRNA-1273/mRNA-1273 | 28 | Yes | 52 | Male |
| Q19 | R.1/mRNA-1273/mRNA-1273 | 21 | Yes | 67 | Female |
| Q21 | R.1/mRNA-1273/mRNA-1273 | >28 | Yes | 57 | Female |
| Q22 | BNT162b2/B.1.526 | *89 | Yes | 42 | Male |
| Q23 | BNT162b2/B.1.526 | *82 | Yes | 32 | Male |
| Q38 | BNT162b2/B.1.1.7 | *59 | Yes | 22 | Female |
| Q39 | BNT162b2/B.1.1.7 | *213 | Yes | 66 | Male |
| Q40 | BNT162b2/B.1.617.2 | *31 | Yes | 50 | Female |
| Q43 | BNT162b2/BNT162b2/B.1.526 | *62 | Yes | 30 | Male |
| Q44 | WA1/mRNA-1273/mRNA-1273 | 114 | Yes | 49 | Female |
| Q45 | WA1/BNT162b2/BNT162b2 | 57 | Yes | 35 | Female |
| Q46 | WA1/BNT162b2/BNT162b2 | 46 | Yes | 30 | Female |
| Q47 | WA1/BNT162b2/BNT162b2 | 57 | Yes | 32 | Female |
| Q48 | WA1/BNT162b2/BNT162b2 | 50 | Yes | 64 | Female |
| Q59 | BNT162b2/BNT162b2/B.1.617.2 | *35 | Yes | 58 | Female |
| Q60 | B.1.617.2/BNT162b2/BNT162b2 | 40 | Yes | 61 | Male |
| Q63 | BNT162b2/BNT162b2/B.1.617.2 | *30 | Yes | 40 | Female |
| Q64 | mRNA-1273/mRNA-1273/B.1.617.2 | *66 | Yes | 29 | Male |
| Q65 | BNT162b2/BNT162b2/B.1.617.2 | *62 | Yes | 33 | Female |
| Q66 | BNT162b2/BNT162b2/B.1.617.2 | *60 | Yes | 42 | Female |
| Q67 | BNT162b2/BNT162b2/B.1.617.2 | *73 | Yes | 37 | Male |
| **BA.1 breakthrough** | | | | | |
| Q24 | BNT162b2/BNT162b2/BA.1 | *14 | Yes | Unknown | Unknown |
| Q25 | BNT162b2/BNT162b2/BA.1 | *14 | Yes | Unknown | Unknown |
| Q26 | mRNA-1273/mRNA-1273/BA.1 | *35 | Yes | Unknown | Unknown |
| Q27 | BNT162b2/BNT162b2/BNT162b2/BA.1 | *135 | Yes | 78 | Male |
| Q28 | BNT162b2/BNT162b2/BNT162b2/BA.1 | *14 | Yes | Unknown | Unknown |
| Q29 | BNT162b2/BNT162b2/BNT162b2/BA.1 | *14 | Yes | Unknown | Unknown |
| Q30 | BNT162b2/BNT162b2/BNT162b2/BA.1 | *14 | Yes | Unknown | Unknown |
| Q31 | BNT162b2/BNT162b2/BNT162b2/BA.1 | *41 | Yes | 48 | Male |
| Q32 | BNT162b2/BNT162b2/BNT162b2/BA.1 | *26 | Yes | 38 | Female |
| Q33 | BNT162b2/BNT162b2/B.1.617.2/BNT162b2/BA.1 | *19 | Yes | 35 | Female |
| Q34 | BNT162b2/BNT162b2/mRNA-1273/mRNA-1273/BA.1 | *67 | Yes | 40 | Male |
| Q41 | WA1/BNT162b2/BA.1 | *21 | Yes | 52 | Male |
| Q42 | WA1/BNT162b2/BA.1 | *44 | Yes | 37 | Intersex |
| **BA.2 breakthrough** | | | | | |
| Q35 | BNT162b2/BNT162b2/BA.2 | *14 | Yes | 50 | Female |
| Q36 | BNT162b2/BNT162b2/BNT162b2/Ad26.COV2.S/BA.2 | *22 | Yes | 69 | Male |
| Q49 | BNT162b2/BNT162b2/mRNA-1273/BA.2 | *16 | Yes | 32 | Male |
| Q50 | mRNA-1273/mRNA-1273/mRNA-1273/BA.2 | *14 | Yes | 34 | Male |
| Q51 | BNT162b2/BNT162b2/mRNA-1273/BA.2 | *19 | Yes | 33 | Female |
| Q52 | BNT162b2/BNT162b2/mRNA-1273/BA.2 | *18 | Yes | 29 | Female |
| Q53 | BNT162b2/BNT162b2/BNT162b2/BA.2 | *25 | Yes | 34 | Male |
| Q54 | BNT162b2/BNT162b2/BNT162b2/BA.2 | *36 | Yes | 37 | Female |
| Q55 | BNT162b2/BNT162b2/mRNA-1273/BA.2 | *18 | Yes | 41 | Female |
| Q56 | mRNA-1273/mRNA-1273/mRNA-1273/BA.2 | *21 | Yes | 36 | Female |
| Q57 | BNT162b2/BNT162b2/mRNA-1273/BA.2 | *32 | Yes | 28 | Male |
| Q58 | BNT162b2/BNT162b2/mRNA-1273/BA.2 | *23 | Yes | 33 | Female |

# Reporting Summary

## Statistics

For all statistical analyses, confirm that the following items are present in the figure legend, table legend, main text, or Methods section.

| n/a | Confirmed | |
|---|---|---|
| ☐ | ☒ | The exact sample size (*n*) for each experimental group/condition, given as a discrete number and unit of measurement |
| ☐ | ☒ | A statement on whether measurements were taken from distinct samples or whether the same sample was measured repeatedly |
| ☐ | ☒ | The statistical test(s) used AND whether they are one- or two-sided *Only common tests should be described solely by name; describe more complex techniques in the Methods section.* |
| ☒ | ☐ | A description of all covariates tested |
| ☒ | ☐ | A description of any assumptions or corrections, such as tests of normality and adjustment for multiple comparisons |
| ☐ | ☒ | A full description of the statistical parameters including central tendency (e.g. means) or other basic estimates (e.g. regression coefficient) AND variation (e.g. standard deviation) or associated estimates of uncertainty (e.g. confidence intervals) |
| ☐ | ☒ | For null hypothesis testing, the test statistic (e.g. *F*, *t*, *r*) with confidence intervals, effect sizes, degrees of freedom and *P* value noted *Give P values as exact values whenever suitable.* |
| ☒ | ☐ | For Bayesian analysis, information on the choice of priors and Markov chain Monte Carlo settings |
| ☒ | ☐ | For hierarchical and complex designs, identification of the appropriate level for tests and full reporting of outcomes |
| ☒ | ☐ | Estimates of effect sizes (e.g. Cohen's *d*, Pearson's *r*), indicating how they were calculated |

*Our web collection on statistics for biologists contains articles on many of the points above.*

## Software and code

Policy information about availability of computer code

| Data collection | SoftMax Pro 7.0.2 (Molecular Devices, LLC) was used to measure luminescence in the pseudovirus neutralization assays. Biacore T200 biosensor (Cytiva) was used to measure the spike-ACE2 binding affinity. |
|---|---|
| Data analysis | GraphPad Prism (version 9.2) was used for data visualization and for statistical tests. PISA was used for indetifying antibody-spike interface residues. PyMOL v.2.3.2 was used to perform mutagenesis and to generate structural plots. SPR data were fitted with Biacore T200 Evaluation Software (Version 1.0). The Racmacs package (https://acorg.github.io/Racmacs/, version 1.1.4) was used to generate the antigenic cartography. |

For manuscripts utilizing custom algorithms or software that are central to the research but not yet described in published literature, software must be made available to editors and reviewers. We strongly encourage code deposition in a community repository (e.g. GitHub). See the Nature Portfolio guidelines for submitting code & software for further information.

## Data

Policy information about availability of data

All manuscripts must include a data availability statement. This statement should provide the following information, where applicable:
- Accession codes, unique identifiers, or web links for publicly available datasets
- A description of any restrictions on data availability
- For clinical datasets or third party data, please ensure that the statement adheres to our policy

All experimental data are provided in the manuscript. Materials used in this study will be available under an appropriated Materials Transfer Agreement. An interactive antigenic map based on the neutralization data of boosted vaccinee sera (Figure 4b) is available online (https://figshare.com/articles/media/OmicronAntigenicMap/19854046). Sequences for Omicron prevalence analysis were downloaded from GISAID (https://www.gisaid.org/). The structures used for analysis in this study are available from PDB under IDs 6ZGE, 7L5B, 6XDG, 7U0N, 7UB0 and 7KMG.

## Human research participants

Policy information about studies involving human research participants and Sex and Gender in Research.

| | |
|---|---|
| Reporting on sex and gender | *Sex and gender of the participants in this study are described in detail in the Extended Data Table 2: 30/63 female, 26/63 male, 1/63 intersex, 6/63 unknown sex; 7/63 unknown age, 56/63 22-78 years old.* |
| Population characteristics | A total of 63 individuals were enrolled in this study. Population characteristics for the sera utilized in the pseudovirus neutralization assays are described in the Extended Data Table 2. |
| Recruitment | Participants volunteered and were enrolled in an observational cohort study at Columbia University Irving Medical Center or at the Hackensack Meridian Center for Discovery and Innovation (CDI) to study the immunological responses to SARS-CoV-2 in individuals who had received COVID-19 vaccines. Self-selection biases may have affected the demographics of the enrolled population, but are not expected to have impacted the results of this study. High titer samples were specifically chosen so that fold-changes in titer could be better determined. |
| Ethics oversight | All collections were conducted under protocols reviewed and approved by the Institutional Review Board of Columbia University or or the Hackensack Meridian Center for Discovery and Innovation. All of the participants provided written informed consent. |

Note that full information on the approval of the study protocol must also be provided in the manuscript.

# Field-specific reporting

Please select the one below that is the best fit for your research. If you are not sure, read the appropriate sections before making your selection.

☒ Life sciences  ☐ Behavioural & social sciences  ☐ Ecological, evolutionary & environmental sciences

For a reference copy of the document with all sections, see nature.com/documents/nr-reporting-summary-flat.pdf

# Life sciences study design

All studies must disclose on these points even when the disclosure is negative.

| | |
|---|---|
| Sample size | No statistical methods were used to predetermine sample size. We used analogous sample sizes as in previous work (e.g. Wang et al 2021, Nature; Liu et al 2022, Nature; Iketani et al 2022, Nature), which we had previously determined to be sufficient sample sizes for comparisons between groups for these experiments. The human research participants (n=63) in this study were characterized in 4 groups, including Boosted (n=16), Non-Omicron infection & vaccination (n=22), BA.1 breakthrough (n=13) and BA.2 breakthrough (n=12). |
| Data exclusions | No data were excluded. |
| Replication | The antibody neutralization assays, the serum neutralization assays, the huACE2 inhibition assays were repeated twice independently in technical triplicate with similar results. SPR assays were repeated twice independently with similar results. The results that are shown are representative. All replicates for the neutralization assays and SPR assays are reproducible and successful. |
| Randomization | As this is an observational study, randomization is not relevant. |
| Blinding | As this is an observational study, investigators were not blinded. |

# Reporting for specific materials, systems and methods

We require information from authors about some types of materials, experimental systems and methods used in many studies. Here, indicate whether each material, system or method listed is relevant to your study. If you are not sure if a list item applies to your research, read the appropriate section before selecting a response.

## Materials & experimental systems

| n/a | Involved in the study |
|-----|----------------------|
| ☐ | ☒ Antibodies |
| ☐ | ☒ Eukaryotic cell lines |
| ☒ | ☐ Palaeontology and archaeology |
| ☒ | ☐ Animals and other organisms |
| ☒ | ☐ Clinical data |
| ☒ | ☐ Dual use research of concern |

## Methods

| n/a | Involved in the study |
|-----|----------------------|
| ☒ | ☐ ChIP-seq |
| ☒ | ☐ Flow cytometry |
| ☒ | ☐ MRI-based neuroimaging |

## Antibodies

| | |
|---|---|
| Antibodies used | All of the antibodies used in this study were produced in our laboratory or provided by other laboratories or companies. 1-20, CAB-A17, LY-CoV555, 2-15, S309, 2-7, LY-CoV1404, ADG-2, DH1047, 10-40, S2X259, 4-18, and 5-7 were expressed and purified in-house as described previously in Liu et al 2020, Nature and in the Methods section of this manuscript. REGN10933, COV2-2196, REGN10987, and COV2-2130 were produced and provided by Regeneron Pharmaceuticals, Brii-196 and Brii-198 were produced and provided by Brii Biosciences, CB6 was produced and provided by Baoshan Zhang and Peter Kwong (NIAID), and ZCB11 was produced and provided by Zhiwei Chen (HKU). |
| Validation | All of the antibodies have been validated in previous studies by neutralization of SARS-CoV-2. Specifically, 1-20, CB6, Brii-196, REGN10933, COV2-2196, LY-CoV555, 2-15, REGN10987, COV2-2130, LY-CoV1404, Brii-198, S309, 2-7, ADG-2, 10-40, S2X259, 4-18, and 5-7 were tested in Liu et al 2022, Nature, Iketani et al 2022, Nature, or Liu et al 2022, Science Translational Medicine. CAB-A17 and ZCB11 were newly produced and tested prior to use in this study and confirmed to have similar results as that of the original publications (Sheward et al 2022, BioRxiv and Zhou et al 2022, BioRxiv, respectively). |

## Eukaryotic cell lines

Policy information about cell lines and Sex and Gender in Research

| | |
|---|---|
| Cell line source(s) | HEK293T cells were obtained from ATCC (Cat #CRL-3216). Vero-E6 cells were obtained from ATCC (Cat #CRL-1586). Expi293 cells were obtained from Thermo Fisher (Cat #A14527). |
| Authentication | Cells were purchased from authenticated vendors and morphology was confirmed visually before use. |
| Mycoplasma contamination | cell lines tested mycoplasma negative. |
| Commonly misidentified lines (See ICLAC register) | No commonly misidentified cell lines were used in this study. |

