## [Peer Review File. · Nature]

Manuscript Title: Antibody evasion by SARS-CoV-2 Omicron subvariants BA.2.12.1, BA.4, and BA.5

Reviewer Comments & Author Rebuttals

Reviewer Reports on the Initial Version:

Referees' comments:

Referee #1 (Remarks to the Author):

Wang and colleagues investigated the antibody sensitivity of BA.2.12.1, BA.4 and BA.5 Omicron subvariants. They assessed the neutralization of monoclonal antibodies and sera from various cohorts of infected or vaccinated individuals using a pseudovirus-based assay. They measured ACE2 affinity of the different spikes by surface plasmon resonance. These results are associated with analyses linking the phenotypes to solved spike structures, or determining antigenic distance between variants. The report is based on a solid methodology and provides timely results about the increased resistance to neutralization of recently circulating Omicron sub-variants.

1. If possible, measuring the neutralizing activity of sera from non-vaccinated individuals infected with Omicron will provide a better understanding of the antigenic distance between Omicron subvariants.
2. The correspondence between antibody numbers and names could be provided on each figure to facilitate reading
3. Lines 69-70. The sentence is confusing as cilgavimab is authorized as part of the Evusheld cocktail.
4. The results with the anti-NTD antibodies (Fig. 2a) are not commented in the text.
5. The "antigenic unit" could be more explicated in the text (line 156). What is the antigenic unit distance for instance between D614G and Delta, or between Delta and Omicron. It has been proposed that Omicron subvarainst may represent a distinct "serotype". This could be discussed
6. The authors should be more cautious when discussing fitness based on ACE2 affinity (for instance line 177). How this parameter correlates with infectivity remains debated. An investigation of spike cleavage efficiency of the variants would be a great addition to this discussion.

Referee #2 (Remarks to the Author):

In this report the investigators describe antigenic variation of the SARS-CoV-2 variants BA..2.12.1 and

BA.4/5 at the monoclonal and serological levels after 3rd-boost vaccination or infection. The report using pseudoviral neutralization assays that while BA.2.12.1 is modestly resistant to sera from vaccinated/boosted subjects, BA.4/5 variants are substantially more resistant. Specific mutations and epitopes classes are described accounting for these variations. Notably, they report that bebtelovimab is the only remaining approved mAb treatment that retains full potency against the BA.2.12.1 and BA.4/5 variants. ACE2 binding activity of the spike proteins from these variants is also compared. While mutations such as F486V that inhibit antigenic class I and II antibodies reduced ACE2 binding a compensatory reversion at R493Q restored affinity. Comments:

1. The main conclusions are made using pseudoviral assays that are useful and clearly identify changes in antigenicity, but actual hot SARS-COV-2 variant isolates could certainly display unique characteristics affecting infectivity and neutralization. The conclusions drawn are valuable but are too strongly stated and generalized based on pseudoviral assays alone. Ideally, real viral isolates should be tested.
2. Again, actual virus isolates in vivo in relevant animal models such as hamsters is the true measure and could identify variations not evident with in vitro pseudoviral assays. For example, viral fitness could well be reduced for BA.4/5 allowing less potent antibody neutralization to suffice for protection. These are important considerations that are not even discussed in this paper.
3. As the field progresses, antigenic site definitions have not become fully standardized such that reference to class I and II epitopes should also include reference to the other nomenclatures (RBD 1, RBD 2, etc).

Referee #3 (Remarks to the Author):

In their manuscript Wang and colleagues characterize neutralization of BA.2.12.1 and BA.4/5 by mAbs and polyclonal serum and also characterize interactions with ACE2. In a way, the manuscript is 'another variant neutralization paper' but it provides important information including regarding therapeutic mAbs. Also, it is well written and the experiments are planned and conducted in a rigorous way.

Major points

- 1) Figure 4d: Showing this in 3D does not allow the reader to really understand differences. This should be broken down to two dimensions as typically done by the Smith group. Also, indicate how an antigenic unit is defined.

Minor points

- 1) Many abbreviations are not defined when first used.
- 2) Line 31: 'The SARS-CoV-2...'
- 3) Line 89 and 92: Why are the amino acid names starting with capitalized letters?
- 4) Line 103: It is unlikely that these >17 mutation in the RBD all contribute to escape from

neutralization. Please rephrase this sentence.

5) Figure 1b: What does the scale bar represent.

6) It would be appropriate to thank GISAID and its contributors for providing the sequence data.

Author Rebuttals to Initial Comments:

Referee #1

Wang and colleagues investigated the antibody sensitivity of BA.2.12.1, BA.4 and BA.5 Omicron subvariants. They assessed the neutralization of monoclonal antibodies and sera from various cohorts of infected or vaccinated individuals using a pseudovirus-based assay. They measured ACE2 affinity of the different spikes by surface plasmon resonance. These results are associated with analyses linking the phenotypes to solved spike structures, or determining antigenic distance between variants. The report is based on a solid methodology and provides timely results about the increased resistance to neutralization of recently circulating Omicron sub-variants.

1. If possible, measuring the neutralizing activity of sera from non-vaccinated individuals infected with Omicron will provide a better understanding of the antigenic distance between Omicron subvariants.

We appreciate the reviewer's suggestion very much, and we would love to have done so. However, it has been prohibitively difficult for us to obtain serum samples from non-vaccinated individuals infected with each Omicron subvariant. Such cases are simply not captured within the patient population seen at our medical center. To date we have only one such sample in our hands, which will not yield meaningful data. In addition, we have asked colleagues in the field for such samples but without success.

2. The correspondence between antibody numbers and names could be provided on each figure to facilitate reading

We have added the clinical name of each monoclonal antibody in the revised Figure 2a, Extended Data Figures 1, 3, and 4.

3. Lines 69-70. The sentence is confusing as cilgavimab is authorized as part of the Evusheld cocktail. We thank the reviewer for pointing this out. We have rephrased the sentence (Lines 59-61 in the revised manuscript) and made it clearer in the revised manuscript.

4. The results with the anti-NTD antibodies (Fig. 2a) are not commented in the text.

We thank the reviewer for the careful reading of the manuscript. The description of the results on the anti-NTD antibodies has been added to the revised manuscript (lines 63-66).

5. The "antigenic unit" could be more explicated in the text (line 156). What is the antigenic unit distance for instance between D614G and Delta, or between Delta and Omicron. It has been proposed that Omicron subvariants may represent a distinct "serotype". This could be discussed.

We have modified the text in lines 155-158 to better explain the antigenic unit. The x-, y- and z-axis represent antigenic distance with one unit in any direction corresponding to one two-fold serum dilution of the neutralizing ID₅₀ titer. Distances from each viral strain to each serum sample correspond to the fold drop in ID₅₀ compared to the maximum titer for each serum. The positions of all points are determined by optimizing all distances to minimize total map error. The map orientation within the x-, y-, and z-axis is free to rotate as only relative distances can be inferred. The distances between circles (viral strains) in the map can be interpreted as a measure of antigenic similarity, where the points positioned more closely

together are antigenically more similar. The distance between D614G and Delta, or between Delta and Omicron, represents the antigenic similarities between these strains as measured by their respective ID₅₀ titers against the same set of sera (for example, boosted sera in this study).

It is hard to tell whether Omicron subvariants represent a distinct 'serotype' or not. Boosted samples (with WT antigens) elicited potent and broadly neutralizing antibodies and showed significantly higher antibody titers against Omicron (Cameroni, E. et al, Nature, 2022; Iketani, S. et al, Nature, 2022; Muik, A. et al, Science, 2022), which suggests Omicron may not be a distinct 'serotype'. In contrast, Omicron infection may not confer broad protection against non-Omicron variants in unvaccinated individuals (Suryawanshi, R. K. et al, Nature, 2022), which supports defining Omicron as a distinct 'serotype'. In our study, we only examined the antigenic cartography for D614G and other Omicron sublineages. Although we see Omicron sublineages are antigenically distant from D614G, we believe additional studies on other VOCs would be needed to answer this question. To keep the Discussion concise, we did not specifically address this point.

6. The authors should be more cautious when discussing fitness based on ACE2 affinity (for instance line 177). How this parameter correlates with infectivity remains debated. An investigation of spike cleavage efficiency of the variants would be a great addition to this discussion.

We again appreciate this comment, and we have modified the text in several places to minimize discussion on fitness, other than the fitness of receptor binding.

Referee #2

In this report the investigators describe antigenic variation of the SARS-CoV-2 variants BA.2.12.1 and BA.4/5 at the monoclonal and serological levels after 3rd-boost vaccination or infection. The report using pseudoviral neutralization assays that while BA2.12.1 is modestly resistant to sera from vaccinated/boosted subjects, BA.4/5 variants are substantially more resistant. Specific mutations and epitopes classes are described accounting for these variations. Notably, they report that bebtelovimab is the only remaining approved mAb treatment that retains full potency against the BA.2.12.1 and BA.4/5 variants. ACE2 binding activity of the spike proteins from these variants is also compared. While mutations such as F486V that inhibit antigenic class I and II antibodies reduced ACE2 binding a compensatory reversion at R493Q restored affinity. Comments:

1. The main conclusions are made using pseudoviral assays that are useful and clearly identify changes in antigenicity, but actual hot SARS-COV-2 variant isolates could certainly display unique characteristics affecting infectivity and neutralization. The conclusions drawn are valuable but are too strongly stated and generalized based on pseudoviral assays alone. Ideally, real viral isolates should be tested.

We followed the reviewer's suggestion and conducted the authentic virus neutralization assays on 16 boosted serum samples and 4 mAbs (COV2-2196 and ZCB11, REGN10987, and LY-CoV1404) to confirm the results generated from pseudoviruses. The new results are shown in the Extended Data Figures 1, 2 and 6 and described in the manuscript text. Overall, the neutralization pattern observed for the boosted sera and the select monoclonal antibodies on authentic viruses is consistent with data generated using pseudoviruses, as we have found in our previous publications on the characterization of variants.

2. Again, actual virus isolates in vivo in relevant animal models such as hamsters is the true measure and could identify variations not evident with in vitro pseudoviral assays. For example, viral fitness

could well be reduced for BA.4/5 allowing less potent antibody neutralization to suffice for protection. These are important considerations that are not even discussed in this paper. We thank the reviewer for this thoughtful comment. We have revised the discussion section accordingly.

3. As the field progresses, antigenic site definitions have not become fully standardized such that reference to class I and II epitopes should also include reference to the other nomenclatures (RBD 1, RBD 2, etc).

We agree and have included the reference on the original classification of RBD-directed monoclonal antibodies. Furthermore, the classification we used in this study is exactly the same as our prior publication in Nature (Wang et al, 2021; Liu et al, Nature 2022; and Iketani et al, Nature 2022). We believe this consistency is useful to readers.

Referee #3

In their manuscript Wang and colleagues characterize neutralization of BA.2.12.1 and BA.4/5 by mAbs and polyclonal serum and also characterize interactions with ACE2. In a way, the manuscript is ‘another variant neutralization paper’ but it provides important information including regarding therapeutic mAbs. Also, it is well written and the experiments are planned and conducted in a rigorous way.

Major points

1) Figure 4d: Showing this in 3D does not allow the reader to really understand differences. This should be broken down to two dimensions as typically done by the Smith group. Also, indicate how an antigenic unit is defined.

We have provided the link to access an active 3D antigenic map (<https://figshare.com/articles/media/OmicronAntigenicMap/19854046>) in Figure 4 legend and “Data availability” section. We think this will allow readers to truly understand the antigenic differences in more detail. We have also included the definition of the antigenic unit in both the revised main text and figure legends. The 2D antigenic map is shown here with similar overall distribution of viruses and sera compared to the 3D map.

Minor points

1) Many abbreviations are not defined when first used.

Abbreviations have been defined in the revised manuscript.

2) Line 31: 'The SARS-CoV-2....'

Revised.

3) Line 89 and 92: Why are the amino acid names starting with capitalized letters?

Revised.

4) Line 103: It is unlikely that these >17 mutation in the RBD all contribute to escape from neutralization. Please rephrase this sentence.

Revised.

5) Figure 1b: What does the scale bar represent.

Added.

6) It would be appropriate to thank GISAID and its contributors for providing the sequence data.

Added.